# Advancements in Synthetic Strategies and Biological Effects of Ciprofloxacin Derivatives: A Review

**DOI:** 10.3390/ijms25094919

**Published:** 2024-04-30

**Authors:** Vuyolwethu Khwaza, Sithenkosi Mlala, Blessing A. Aderibigbe

**Affiliations:** Department of Chemistry, University of Fort Hare, Alice Campus, Alice 5700, South Africa; smlala@ufh.ac.za

**Keywords:** ciprofloxacin, derivatives, synthetic strategies, biological activities

## Abstract

Ciprofloxacin is a widely used antibiotic in the fluoroquinolone class. It is widely acknowledged by various researchers worldwide, and it has been documented to have a broad range of other pharmacological activities, such as anticancer, antiviral, antimalarial activities, etc. Researchers have been exploring the synthesis of ciprofloxacin derivatives with enhanced biological activities or tailored capability to target specific pathogens. The various biological activities of some of the most potent and promising ciprofloxacin derivatives, as well as the synthetic strategies used to develop them, are thoroughly reviewed in this paper. Modification of ciprofloxacin via 4-oxo-3-carboxylic acid resulted in derivatives with reduced efficacy against bacterial strains. Hybrid molecules containing ciprofloxacin scaffolds displayed promising biological effects. The current review paper provides reported findings on the development of novel ciprofloxacin-based molecules with enhanced potency and intended therapeutic activities which will be of great interest to medicinal chemists.

## 1. Introduction

Ciprofloxacin (**1**, Figure 1) is among the top five most commonly manufactured generic antibiotics worldwide and is classified within the quinolone antibiotic group [1,2,3]. It is one of the most widely used drugs for the treatment of various bacterial infections that affect the urinary tract, the respiratory system, the skin, the gastrointestinal system, the abdominal region, bones, and joints [4,5,6,7,8,9]. Fluoroquinolones exhibit good tolerability, boasting a strong safety profile and advantageous pharmacokinetic characteristics alongside their wide range of antibacterial activities [10,11]. They are one of the primary classes of pharmacophores playing a pivotal role in drug discovery and development. The discovery of fluoroquinolone-based drugs has garnered significant interest due to their extensive array of pharmacological effects [12]. Ciprofloxacin can reduce bacterial antibiotic resistance when combined in altered concentrations with other antibiotics [13,14]. Its antibacterial effect is via inhibition of type II bacterial topoisomerase enzymes, including DNA gyrase, an enzyme that is involved in DNA replication, recombination, and repair [15]. In addition, it can bind to topoisomerase IV. Topoisomerase protein–DNA complexes are trapped, causing DNA damage, inducing cell death mechanisms, and interfering with normal DNA replication [16].

The diverse pharmacological effects of ciprofloxacin have been explored in several studies, including its potential anti-TB [17,18,19,20], antifungal [21,22,23], antiviral [24,25,26,27], antimalarial [28], and antitumor activities [29]. Due to its exceptional pharmacological activities, the synthesis of ciprofloxacin derivatives has recently received extensive attention [30]. The development of new drugs with promising clinical applications and novel mechanisms of action is difficult, expensive, and time-consuming. Numerous researchers employ a budget-friendly strategy to enhance therapeutic effectiveness by modifying existing drugs rather than pursuing the expensive route of developing new therapeutics and subjecting them to clinical testing. The foundational method for drug discovery involves employing structure–activity relationships of lead compounds, like already existing drugs, to produce analogues that offer enhanced therapeutic properties [31,32]. Ciprofloxacin derivatives have demonstrated a range of pharmacological potentials, including anticancer, antibacterial, antimicrobial, antimalarial, and antiviral properties [4,33,34,35,36]. In 2013, Castro et al. [37] reported several ciprofloxacin derivatives with potential medicinal applications. The current review focuses on recent advancements in synthetic strategies and potential pharmacological activities of ciprofloxacin derivatives.

## 2. Modification of Ciprofloxacin Structure

As illustrated in Figure 1, the chemical structure of ciprofloxacin has a central core consisting of a quinolone nucleus, a fluorine atom present at position C-6, and a piperazine ring attached at position C-7. Researchers have used different synthetic methods to modify the chemical structure of ciprofloxacin to develop derivatives with enhanced antimicrobial properties. Figure 1 illustrates several possible modifications to the structure of Ciprofloxacin. This can involve changes to the core structure of the molecule or the addition of functional groups to improve its activity against specific microbial pathogens. However, most studies have concentrated on modifying two major active sites in the ciprofloxacin structure, specifically the carboxylic acid group at position C-3 and the piperazine group at position C-7. The two positions are common sites for modifications because modifications here can significantly affect the drug’s pharmacological properties, as demonstrated in Table 1. Typically, an aromatic ring or other substituents are added to these positions.

The development of ciprofloxacin derivatives has yielded molecules with a broad spectrum of pharmacological activities and minimal toxic or side effects, resulting from molecular modification for lead optimization through bioisosteric replacements, homologation of the side chain or branching of the side chain, stereochemistry, and other helpful techniques for derivative design and development. Ciprofloxacin derivatives with increased potency and activity have been reported in several publications. It has been discovered that the addition of distinct groups at position C-7 of the quinolone nucleus influences the quinolone’s potency, bioavailability, and physicochemical characteristics, in addition to its affinity for DNA gyrase (a protein found in Gram-negative bacteria) and/or Topoisomerase IV (a protein found in Gram-positive bacteria) [38,39]. Furthermore, adding an aryl group to the fluoroquinolone’s piperazine moiety changed its antibacterial activity to antiviral and anticancer activity [39]. Some studies indicate that the central 4-oxo-3-carboxylic acid structure serves as the primary binding site for DNA-gyrase, and modifying this fragment resulted in a notable reduction in antibacterial effects [4,6,37]. The future prospects for ciprofloxacin derivatives look promising, with research focusing on novel synthesis strategies that aim to enhance their biological potency and address resistance issues. Ciprofloxacin, a widely used fluoroquinolone antibiotic, has been instrumental in treating various infections. However, the growing concern of drug resistance has driven the need for innovative approaches to develop more effective derivatives. The following section expands upon the previously discussed literature concerning modifications made to ciprofloxacin’s structure that have resulted in significantly improved biological activities.

**Table 1 ijms-25-04919-t001:** The outcomes of structural modification made to Ciprofloxacin at the C-3 and C-7 positions.

Type of Modifications	Outcomes	Bibliography
Modification of the C-3 carboxylic acid group	Induced apoptosis in cancerous estrogen-negative cells (MDA-MB-231)	[40]
	Anti-MRSA activity decreased dramatically (MIC: >201 μg/mL)	[41]
	The introduction of an ester group (methyl, ethyl, or propyl ester) in place of the original carboxyl group of ciprofloxacin may have greater penetration and higher efficacy in the treatment of bacterial infections of skin and soft tissues and consequently have substantial potential for the clinical treatment of MRSA and MSSA infection	[42]
	Substitution of the carboxylic group of the quinolones by 1,2,4-triazolyl, 1,3,4-oxadiazolyl, a-amino ester or hydrazide derivatives of aldehyde sugars do not influence the antibacterial activity	[43]
Modification of the C-7 position	Has shown remarkably broad cytotoxicity against cancer cells HT29, HCT116, and SW620, rivaling or even surpassing the effect of cisplatin. It has the potential to enhance obesity treatment by inhibiting fat absorption throughout the gastrointestinal tract	[44]
	Suppression of SARS-CoV-2 viral replication and the virus’s M proenzyme both in vitro and in silico in a dose-dependent manner	[24]
	Displays significant cytotoxic activity against U-87 and MCF-7 cancer cell lines	[45]
	Enhances the antibacterial activity and influences the interaction with the enzyme DNA gyrase	[46]
	Showed high antiproliferative activities against prostate PC3 cancer cells	[47]
	Demonstrated comparable or superior antibacterial activity compared to ciprofloxacin against *A. baumannii* and *B. cereus* 138^®^	[48]

## 3. Biological Activities of Ciprofloxacin Derivatives

### 3.1. Antibacterial Activity

Most hospital-acquired infections are caused by bacterial infections, which pose a serious threat to global health and significantly increase the cost of healthcare systems worldwide by causing high rates of mortality [49,50]. The evolution of highly pathogenic strains, such as drug-resistant pathogens with varying degrees of resistance, like vancomycin-resistant *Enterococcus* (VRE), methicillin-resistant *Staphylococcus aureus* (MRSA), and *Escherichia coli*, has already surfaced and has been linked to high death rates [51]. Fluoroquinolone antibiotics like ciprofloxacin are used as alternatives to vancomycin for the treatment of MRSA infections, particularly oral infections [52,53]. Unfortunately, when ciprofloxacin is administered to treat MRSA strains, these bacterial pathogens promptly acquire resistance to the antibiotic [54]. Research indicates that more than 89% of MRSA strains exhibit resistance to ciprofloxacin, with the prevalence of ciprofloxacin-resistant and methicillin-resistant *S. aureus* (CR-MRSA) strains on the rise [55]. New and exceptionally potent molecules that demonstrate strong efficacy against infections, whether they are susceptible to drugs or resistant, are urgently required. To find novel antibiotics, several ciprofloxacin derivatives have been developed. Zhang et al. [4] thoroughly documented the synthesis of ciprofloxacin derivatives with potential antibacterial activities until 2018. Jia et al. [56] provided an update between 2018 and 2021 on the most recent developments of fluoroquinolone derivatives with potential antibacterial properties. This section provides an update on the synthetic strategies of ciprofloxacin derivatives with potential antibacterial activities reported between 2021 and 2024.

Yang et al. [57] effectively synthesized twelve new fluoroquinolone derivatives by attaching *N*-acylarylhydrazone to ciprofloxacin at the C-7 site and evaluated their ability to inhibit the growth of selected Gram-negative and Gram-positive bacteria. The reactions involved heating solutions of ciprofloxacin and ethyl chloroacetate in dimethylformamide (DMF) in alkaline conditions, resulting in the formation of compound **2** with an 85% yield (Figure 2). Compound **2** was then treated with hydrazine hydrate under high-temperature conditions, yielding compound **3** with an 88% yield. The synthesis of the highly active hybrids **4** and **5**, with respective yields of 87% and 42%, was achieved by reacting compound **3** with selected aldehydes in ethanol. The antibacterial assessment revealed that incorporating acyl hydrazone derivatives at the C-7 positions of ciprofloxacin yielded compounds with superior efficacy compared to ciprofloxacin against *S. aureus*. Compound **6** demonstrated a 2-fold improvement in activity compared to ciprofloxacin, exhibiting an MIC value of 0.25 μg/mL against *S. aureus*, while compound **5** displayed a 4-fold enhancement with an MIC value of ≤0.125 μg/mL. Notably, compound 4 exhibited the highest potency against the tested strains, with MIC values of 1 μg/mL against *S. aureus* and *E. coli* and 16 μg/mL against *Pseudomonas aeruginosa*. Additionally, it demonstrated anti-MRSA activity, with a MIC value of 32 μg/mL. Time–killing assays confirmed compound **4** is a promising candidate, and maintains the rapid bactericidal action of ciprofloxacin against *E. coli* (within 2 h) and *S. aureus* (within 4 h). Furthermore, cytotoxicity and hemolysis tests showed low toxicity of the compounds in vitro. Compound **4** also exhibited a high affinity for topoisomerase IV, with a binding energy of −9.9 kcal/mol, surpassing that of ciprofloxacin, according to molecular docking studies.

Using hybridization techniques, Osman et al. [58] devised novel ciprofloxacin–1,3,4-thiadiazole hybrids and then assessed their antimicrobial efficacy through screening of the synthesized compounds against *Klebsiella pneumonia* isolates. Their analysis identified compound **7**, characterized by its MIC value, as the most promising within the series, displaying good sensitivity across all tested strains. The highly potent compound **7** was synthesized following the procedure outlined in Figure 3. Significantly, this study identified compound **7** as pivotal to the project, as it had previously been documented in reactions involving ciprofloxacin and thiosemicarbazide in the presence of phosphorous oxychloride [59,60] and concentrated sulfuric acid [61] and in various other techniques [62]. Additionally, an alternative method was reported in the literature [63], which primarily substitutes phosphorous oxychloride with concentrated sulfuric acid to effectively accomplish the cyclization step. Ciprofloxacin was combined with thiosemicarbazide in concentrated sulfuric acid, followed by refluxing the mixture for 5 h in a water bath. The resultant mixture was subsequently cooled to room temperature, diluted with 10 mL of cold water, and alkalized using diluted ammonia, resulting in the formation of a precipitate, which was rinsed with water and recrystallized from ethanol. Compound **7** was obtained with a satisfactory yield of 67%. Subsequently, the authors conducted a comprehensive investigation of the compound **7** profile, including MBC, time-to-kill assays, and resistance development. The results indicate that while hybridization prolonged the action of compound **7** and reduced drug resistance compared to the parent drug, ciprofloxacin, it also delayed its onset due to increased lipophilicity. Furthermore, molecular docking suggested that the inclusion of a thiadiazole ring facilitated interactions with DNA gyrase similar to those of the parent ciprofloxacin. Lastly, in silico physicochemical and pharmacokinetic studies predicted compound **7** as a promising antibiotic.

Ibrahim and co-authors [64] used the hybridization strategy to develop various ciprofloxacin–sulfonamide hybrid molecules based on six sulfonamides (sulfaguanidine, sulfaclozine, sulfaquinoxaline, sulfadiazine, sulfanilamide, sulfaclozine, and sulfamethoxazole). The authors developed ciprofloxacin–sulfonamide hybrids with different linkers. The methods used to produce these hybrids are illustrated in Figure 4. The initial set, represented by compounds **9** and **10**, used a diazenyl linker and was synthesized by reacting ciprofloxacin with the diazonium salt of specific sulfonamides (**8**) at a pH of 6. Subsequently, compounds **12**–**15**, and **18**, which employed acetamide, 3-propanamide, and 2-propanamide linkers, respectively, were synthesized following the schemes outlined in Figure 4. Initially, these compounds were synthesized by acylating appropriate sulfonamides **8** with corresponding chloroacyl chloride in dry DMF. The resultant acyl derivatives **11**, **16**, and **17** were obtained with high yields (81.4–87.57%) under mild conditions, at room temperature without the necessity of a catalyst. These acyl derivatives were then subjected to reaction with ciprofloxacin in DMF, using reflux heating in the presence of triethylamine (TEA) as a basic catalyst. The compounds were assessed for their ability to inhibit DNA topoisomerase IV and DNA gyrase in vitro. Five ciprofloxacin–sulfonamide hybrids (**12**–**15** and **18**) demonstrated strong inhibitory effects against DNA topoisomerase IV, with IC_50_ values ranging from 0.23 to 0.44 µM. Similarly, three hybrids (**12**–**14**) efficiently inhibited DNA gyrase, with IC_50_ values ranging from 0.43 to 1.1 µM. Moreover, all synthesized hybrids were tested for their MICs against both Gram-positive and Gram-negative pathogens, including *S. aureus* Newman and *E. coli* ATCC8739. Compounds **9** and **10** exhibited significantly improved antibacterial activity compared to ciprofloxacin, with MIC values of 0.324 and 0.422 µM against *S. aureus* Newman and 0.013 µM against *E. coli* ATCC8739. Subsequently, compounds **9**, **12**, **14**, and **18**, identified as potent inhibitors, were evaluated in vivo for CNS side effects, showing no mortality or reduced incidence of convulsions compared to the reference drug. Western blot tests indicated lower GABA (the neurotransmitter Gamma-Aminobutyric Acid) expression for these compounds compared to ciprofloxacin, suggesting minimal CNS side effects. Additionally, a molecular docking study revealed that the hybrid compounds bind similarly to ciprofloxacin, with the sulfonamide moiety occupying an extended pocket for the bulky C-7 substituent.

A ciprofloxacin hybrid, **22**, was designed and synthesized by Elgedamy et al. [65] by acetylating/alkylating ciprofloxacin and then thioamidation on its matching amide, **21**. The desired hybrid molecules were synthesized according to the procedure described in Figure 5. Initially, the acetylated amine **20** was synthesized through a biphasic reaction involving bromoacetyl bromide. Subsequent refluxing of **20** with ciprofloxacin in a basic environment resulted in the production of amide **21** with a favorable yield (54.50%). Thioamidation of **21** was achieved by utilizing phosphorus pentasulfide in dioxane, yielding a 40.50% moderate yield of thioamide **22**. Both compounds **21** and **22** underwent screening to assess their antibacterial effects against various bacterial strains, including *S. aureus*, MRSA, and *E. coli*. Additionally, their antifungal properties were tested against *Candida albicans*. The amide derivative exhibited potent antibacterial efficacy against all the tested strains compared to the original antibiotic, ciprofloxacin. Conversely, thioamide demonstrated robust antifungal activity and moderate antibacterial effects against Gram-negative bacteria (*E. coli*) and Gram-positive bacteria (*S. aureus*, MRSA) when compared to ciprofloxacin. The introduction of a bulky functional group at the C-7 position of ciprofloxacin significantly affected its effectiveness against bacteria, its spectrum, and its safety profile. The addition of an amide group primarily enhanced the antibacterial properties, and the incorporation of a thioamide group predominantly enhanced the antifungal activity, although it also altered the antibacterial efficacy against certain bacterial strains. This suggests that the release of H_2_S plays a more substantial role in enhancing antifungal effects compared to antibacterial effects.

Al-Wahaibi et al. [66] developed a novel class of ciprofloxacin hybrid molecules made up of different heterocycle derivatives, which were tested in vitro against both Gram-negative and Gram-positive bacterial strains, such as *E. coli* and *P. aeruginosa*, as well as *S. aureus* and *B. subtilis*. Compounds **23**–**27**, which are oxadiazole derivatives, exhibited antibacterial effectiveness ranging from 88% to 120% compared to ciprofloxacin against both Gram-positive and Gram-negative bacteria. It is notable that compound **27** exhibited greater efficacy (120%) against *S. aureus* compared to ciprofloxacin. Oxadiazoles **25** and **26** demonstrated comparable effectiveness to ciprofloxacin against both *S. aureus* and *E. coli*. Based on prior research results, incorporating oxadiazole and thiadiazole groups into a compound provided enhanced effectiveness. The synthetic strategy for synthesizing compounds **23**–**27** is detailed in Figure 6, and the previously mentioned intermediates **a**–**e** were produced as described in previous studies [67]. As depicted in Figure 6, ciprofloxacin reacts through a Mannich reaction with heterocycles **a**–**e** and formaldehyde in ethanol at reflux temperature, producing the desired compounds **23**–**27** in good yields varying from 85 to 96%.

Alsughayer et al. [68] developed novel derivatives of ciprofloxacin using various organic reagents and evaluated their efficacy against diverse microorganisms. Ciprofloxacin underwent acylation via a reaction with ethyl cyanoacetate and ethyl acetoacetate under basic conditions, yielding the cyanacetylpiprazinyl dihydroquinoline derivative **28** and the oxobutanoylpiprazinyl dihydroquinoline derivative **29**, respectively. Conversely, *N*-alkylated derivatives **30**–**32** were synthesized by reacting ciprofloxacin with chloroacetonitrile, chloroacetyl acetone, and chloroacetone in the presence of carbonate salt, as illustrated in Figure 7. Compounds **28**–**32** demonstrated notable efficacy against *E. coli* (ATCC 8739). Overall, their efficacy surpassed that of ciprofloxacin discs (5 μg, Oxoid.com), indicating the impact of the modifications made at C-7. Compound **30** exhibited the most significant effect, followed by **29**–**32** (which exhibited similar effects), and, finally, Compound **28**. Similarly, the impact of compound **30** on *S. aureus* (ATCC 25923) was highly effective, followed by compounds **28**, **30**, and **31** (which exhibited similar effects). All these derivatives displayed significant efficacy against *S. aureus* (ATCC 25923), surpassing the effects of the commercial ciprofloxacin disc (5 μg) sample. Moreover, compound **30** exhibited potent activity against *P. aeruginosa* (ATCC 27853). The efficacies of the modified ciprofloxacin derivatives against *P. aeruginosa* (ATCC 27853) can be ranked as follows: **30** > **28** > **32** > **29** > Ciprofloxacin disc > **31.** For *P. aeruginosa* ATCC (27853) and *K. pneumoniae* ATCC 13853, compound **28** exhibited a significant effect, following compound **30**; this could have occurred due to the formation of a quaternary ammonium salt. This finding aligns with findings documented in a recent study [69].

The research carried out by Szostek et al. [70] reported ciprofloxacin derivatives containing menthol and thymol scaffolds (Figure 8), subsequently evaluating them for their antimicrobial properties. These compounds were synthesized using various carboxylic linkers. They believed that linking molecules with antimicrobial properties, distinct from ciprofloxacin in their modes of action, could potentially enhance pharmacological properties or contribute to reducing bacterial resistance. Modifications made to the reaction protocol enabled the synthesis of three variations of derivatives containing a chain-extending linker connected to the piperazinyl moiety (C-7) of ciprofloxacin, ranging from acetyl to hexyl substitutions. Alongside the target primary compounds, disubstituted derivatives **38**, **39**, **46**, and **47** were obtained. For instance, compound **37** was obtained as the primary product with an 85% yield, while a secondary product, compound **39**, resulting from a concurrent reaction with the ciprofloxacin carboxylic group, was isolated with an 8% yield. Amongst these new ciprofloxacin derivatives, compound **43**, which was hybridized with thymol, was significantly effective against various Gram-positive *Staphylococci* strains, with MIC values between 0.8 and 1.6 µg/mL. It was identified as the most potent compound, with MIC values of 1 µg/mL for clinical strains of *Staphylococcus pasteuri* KR 4358 and *S. aureus* (T 5591). All compounds with single substitutions displayed a wide range and potent spectrum of activity. Modification via the 4-oxo-3-carboxylic acid core of ciprofloxacin led to a reduction in its antimicrobial effectiveness.

Lukin et al. [48] investigated the potential use of spirocyclic piperidines for developing derivatives of ciprofloxacin. By employing a halogen-substituted fluoroquinolone core, 36 new ciprofloxacin derivatives were synthesized and evaluated against a combination of two types of Gram-positive bacteria and three types of Gram-negative bacteria. Among these derivatives, compound **53** (Figure 9) stood out as the one showing activity against all five strains. This compound was synthesized by esterification of a commercially accessible 7-chloro-1-cyclopropyl-6-fluoro-4-oxo-1,4-dihydroquinoline-3-carboxylic acid (**48**) to produce ester **49**. This ester was then transformed into boron chelation complex **50** using established methodologies [71,72,73], wherein the chlorine at position C-7 displayed enhanced reactivity in nucleophilic aromatic substitution reactions. Heating compound **50** with spirocyclic piperidines (**51**) at 60 °C alongside TEA prompted the desired substitution. The boron chelation complex was subsequently eliminated by treating intermediates (**52**) with a 2% aqueous sodium hydroxide solution, resulting in the production of fluoroquinolones (**53**) in yields ranging from moderate to nearly complete. The antibacterial effects of the newly synthesized spirocyclic compounds were notably influenced by the peripheral groups in 1-oxa-9-azaspiro. Overall, this new series of compounds demonstrated significant activity against two specific strains: the Gram-negative *Acinetobacter baumannii* 987^®^ and the Gram-positive *Bacillus cereus* 138^®^. Some of these compounds exhibited equal or greater potency compared to ciprofloxacin against these strains.

Pedrood and colleagues [74] synthesized a novel series of *N*-thioacylated derivatives of ciprofloxacin using a Willgerodt–Kindler-type reaction at a moderate temperature without the need for a catalyst. The reaction outlined in Figure 10 illustrates the synthetic strategy used to synthesize *N*-thioacylated ciprofloxacin derivatives **56**–**57**. Commercial ciprofloxacin underwent thioamidation with different aldehydes (**54**–**55**) in the presence of sulfur (S8) in equivalent amounts, resulting in the respective end products **56**–**57**. Initially, the secondary amine group of ciprofloxacin underwent a nucleophilic attack on elemental cyclooctasulfur (S8), leading to cleavage of the S–S bond and the reversible formation of polysulfide anion II. Concurrently, various aldehydes (**54**–**55**) reacted with another molecule of ciprofloxacin to generate the intermediate iminium ion III by eliminating the hydroxyl group. The interaction between intermediates II and III then resulted in the formation of IV, which was subsequently oxidized to produce products **56**–**57**.

The antibacterial effectiveness of these compounds was evaluated in vitro against four bacterial strains: *E. coli*, *P. aeruginosa*, *S. aureus*, and *Staphylococcus epidermidis*. Several compounds exhibited enhanced activity against *S. epidermidis* compared to the parent drug, ciprofloxacin. Some of them demonstrated comparable efficacy to ciprofloxacin against *S. aureus*. Notably, compound **57** showed promising activity against all tested bacteria, with its effectiveness against *S. epidermidis* being twice that of ciprofloxacin, while its performance against *S. aureus* was comparable to that of the parent drug. Moreover, compound **57** displayed similar effectiveness to ciprofloxacin against *P. aeruginosa* and *E. coli*. Additionally, these compounds demonstrated superior inhibitory activity against the JB urease enzyme compared to standard inhibitors such as hydroxyurea and thiourea.

Struga and colleagues [47] detailed the development of new ciprofloxacin derivatives (**58**–**78**) by altering the C-7 piperazinyl group with halogenated acyl chlorides. These compounds were tested for their ability to inhibit the growth of both Gram-positive and Gram-negative bacteria in vitro. The objective of the study was to develop a diverse range of amide derivatives of ciprofloxacin by connecting its C-7 piperazinyl group with halogenated acyl chlorides varying in hydrocarbon structure and halogen type. The compounds were synthesized via a one-step reaction conducted under mild conditions (Figure 11). This involved adding acyl chloride to ciprofloxacin and a suspension of triethylamine in dichloromethane (DCM) at 2−5 °C, followed by stirring the resulting solution at room temperature for 3 h. The relatively low yields of short-chain amides **60** and **65** (28 and 34%, respectively) were attributed to their limited solubility in the solvent and the challenges associated with separation and purification. Conversely, derivatives with longer aliphatic chains (**61**–**64** and **67**–**70**) were obtained with better yields ranging from 41% to 86%.

Among these, conjugates **58** and **59** exhibited significantly enhanced potency compared to ciprofloxacin against various *Staphylococci* strains, with MICs ranging from 0.05 to 0.4 μg/mL. Several derivatives containing chloro (**60**–**64**), bromo (**65**–**68**), and CF_3_-alkanoyl (**71**–**73**) moieties demonstrated either similar or superior activity to ciprofloxacin against certain Gram-positive strains. Notably, some ciprofloxacin analogues (**62**, **67**, and **68**) showed increased effectiveness against selected clinical Gram-negative bacteria. Conjugates **62**, **67**, and **68** also displayed significant impacts on the bacterial growth cycle over 18 h. Moreover, compounds **59**, **61**–**64**, **66**–**69**, and **78** exhibited potent ionostatic activity against three *Mycobacterium tuberculosis* isolates, surpassing the efficacy of conventional antitubercular drugs. Amides **58**, **59**, **62**, **63**, **67**, and **68** targeted enzymes and topoisomerase IV at concentrations ranging from 2.7 to 10.0 μg/mL, indicating that the mechanism of antibacterial action is similar to ciprofloxacin. Molecular docking studies further supported these findings.

Aziz and colleagues [75] synthesized a series of hybrids combining ciprofloxacin with thiazolidine-2,4-dione. The compounds **87**–**93** were prepared according to the procedure outlined in Figure 12. Thiazolidine-2,4-dione was synthesized by heating chloroacetic acid and thiourea, as previously described [76]. Acylated ciprofloxacin was obtained by reacting ciprofloxacin with bromoacetyl bromide in the presence of potassium carbonate [77]. Derivatives **80**–**85** of 5-benzylidene thiazolidine-2,4-dione were synthesized via Knoevenagel condensation of different aromatic aldehydes in glacial acetic acid with anhydrous sodium acetate as a catalyst [78]. The target compounds **87**–**93** were then prepared by alkylating compounds **80**–**85** with acylated ciprofloxacin **1** in DMF with potassium hydroxide (KOH) as the base. The evaluation of their antibacterial effects demonstrated a shift towards combating Gram-positive bacteria. Compounds **87**–**93** displayed significant activity against *S. aureus* ATCC 6538, with MIC values ranging from 0.02 to 0.36 µM, compared to ciprofloxacin (MIC = 5.49 µM). These compounds also exhibited promising activity against MRSA AUMC 261, with **87**, **88**, and **92** displaying MIC values of 5 nM. However, their effectiveness against Gram-negative bacteria was reduced, although compound **92** showed a slightly enhanced activity against *K. pneumoniae* ATCC10031 (MIC = 0.08 µM). The incorporation of the thiazolidine-2,4-dione ring into ciprofloxacin preserved its capability to inhibit DNA synthesis by targeting both topoisomerase IV and the DNA gyrase of *S. aureus*. Compounds **87**, **92**, and **93** exhibited greater potency than ciprofloxacin in inhibiting topoisomerase IV (IC_50_ = 0.3–1.9 μM) and gyrase (IC_50_ = 0.22–0.31 µM), which correlated with their antibacterial activity against *S. aureus* ATCC 6538. Molecular docking against the DNA gyrase active site confirmed the compounds’ ability to form stable complexes with the enzyme, suggesting that **87**, **90**, **91**, **93**, and **92** represent promising broad-spectrum antibacterial agents with good capability to target topoisomerase IV and gyrase enzymes to produce notable efficacy against MRSA.

Abdel-Rahman and colleagues [79] developed Mannich bases derived from ciprofloxacin. These Mannich bases were synthesized by refluxing ciprofloxacin with suitable phenolic compound in excess formaldehyde dissolved in ethanol, resulting in good yields (Figure 13).

Within this group, compounds **94** and **95** displayed remarkable effectiveness against both Gram-positive and Gram-negative bacteria. Notably, compound **98** exhibited significant activity against *E. coli* and *P. aeruginosa*, with MIC values of 0.036 and 0.043, respectively. Compound **96** demonstrated a considerable increase in activity, showing 27-fold and 22-fold enhancements over ciprofloxacin against *S. aureus* and MRSA reference strains. Furthermore, compound **96** displayed potent activity against *S. aureus*, MRSA (reference strain), and MRSA (clinical strain) with MIC values of 0.57, 0.52, and 0.082 µg/mL, respectively. Intriguingly, the most potent compounds exhibited favorable physicochemical properties and drug-likeness characteristics. Mannich bases **98**, **96**, and **97** emerged as promising antibacterial compounds.

Qin and colleagues [18] utilized a hybrid strategy to integrate *N*-(amino)piperazine components from rifamycins into the ciprofloxacin core. Various substitutions were applied to *N*-(amino)piperazine to investigate their potential impact as anti-Tuberculosis (anti-TB) and antibacterial agents. Figure 14A depicts the synthetic pathway for synthesizing compounds **104**–**105.** Starting with the quinolone core compound **99**, treatment with boric acid and acetic anhydride produced boric chelate **100** using established methods. Chelate **100** was then coupled with compound **105** in the presence of Et_3_N, and subsequent removal of the Boc group resulted in the intermediate, **103.** The condensation of **103** with various aldehydes resulted in the synthesis of target compounds **104** and **105**. Figure 14B outlines the detailed synthetic pathway for obtaining target compound **111.** Initially, Buchwald–Hartwig cross-coupling of readily available compound **106** with 1-bromo-4-(trifluoromethoxy)benzene (**107**) yielded compound **108**. The subsequent introduction of a methyl group to the amino moiety via reductive amination resulted in compound **109**. Treatment of **109** with trifluoroacetic acid (TFA) led to the removal of the Boc group, generating compound **110**. Finally, the coupling of compound **110** with quinolone boric chelate **100** yielded the target compound, **111**.

Compounds **105**, containing a cyclohexyl group (MIC = 0.28 μM), and **111**, with an *N*-methyl-*N*-isopropyl group (MIC = 0.60 μM), demonstrated anti-TB activity comparable to ciprofloxacin (MIC = 0.34 μM). Compound **104** (MIC = 2 to 8 μg/mL) exhibited significantly higher potency (4 ≥ 32-fold) against MRSA and *VRE Enterococcus faecium* 18-4 compared to ciprofloxacin (MIC =16 ≥ 64 μM). Currently, compounds **104** and **105** are being pursued as potential leads for further refinement in antibacterial and anti-TB drug development, respectively.

Mohammed et al. [80] synthesized a series of new compounds derived from ciprofloxacin, incorporating *N*-4 piperazinyl groups as urea links with chalcone and thioacetyl groups as pyrimidine links. The *N*-4-piperazinyl ciprofloxacin–chalcones **113**–**115** were synthesized according to previously described methods [81]. Compound **112** was condensed with different aromatic aldehydes in ethanol using a 60% solution of sodium hydroxide, resulting in the desired hybrids **113**–**115** with good yields (Figure 15A). The intended intermediate pyrimidine derivatives, **116**–**119**, were synthesized following a previously described method [82,83], which involved heating a solution of ethyl cyanoacetate, thiourea, and the appropriate aromatic aldehyde in absolute ethanol under reflux conditions, with potassium carbonate serving as a base. Preparation of the ciprofloxacin derivative **120** involved the treatment of ciprofloxacin with bromoacetyl bromide in dichloromethane at 0 °C, with TEA present [75]. Subsequent alkylation of pyrimidine derivatives **116-119** with compound **120** was accomplished in acetonitrile in the presence of TEA, yielding the desired new compounds **121**–**124** (Figure 15B).

These compounds were tested for their capability to combat bacterial and fungal strains, such as *S. aureus*, *P. aeruginosa*, *E. coli*, and *C. albicans*. Results showed that these compounds displayed broad antibacterial activity against both Gram-positive and Gram-negative bacteria, with MICs ranging from 0.06 to 42.23 µM, compared to ciprofloxacin’s MIC range of 0.15 to 3.25 µM. Compounds **114**, **115**, **and 121**–**124** exhibited potent antibacterial activity, with MIC values ranging from 0.06 to 1.53 µM. Additionally, some compounds showed antifungal activity against *C. albicans* comparable to that of ketoconazole, with MIC values ranging from 2.03 to 3.89 µM. Further investigation revealed that certain ciprofloxacin hybrids inhibited DNA gyrase, a potential molecular target, with IC_50_ values ranging from 0.231 ± 0.01 to 7.592 ± 0.40 µM. Docking studies confirmed the capability of compounds **118**, **119**, **122**, **126**, and **124** to form stable complexes with the active site of DNA gyrase (PDB: 2XCT), similar to ciprofloxacin.

Cardoso-Ortiz et al. [36] synthesized a total of twelve hybrid molecules by introducing tetrazoles at position C-7 of the fluoroquinolone scaffolds, ciprofloxacin, and norfloxacin. Figure 16 illustrates the synthetic pathways of ciprofloxacin or norfloxacin–tetrazole hybrids. The typical approach for producing tetrazoles involves a 1,3-dipolar cycloaddition, where an azide (such as sodium azide, hydrazoic acid, or trimethylsilyl azide) reacts with imidoyl chlorides, amides, thioamides, nitriles, isocyanates, and ketene imines as initial compounds. Brönsted or Lewis acids were employed to trigger the activation of the substrates; alternatively, phase-transfer conditions were utilized. The Ugi-azide reaction has been utilized to synthesize 1,5-disubstituted tetrazoles, incorporating four components concurrently: an aldehyde or ketone, an amine, trimethylsilyl azide, and an isocyanide for synthesis. This broad, succinct, innovative approach can additionally use aldo/keto acids/esters within the Ugi-azide reaction to gain access to numerous novel scaffolds.

Among these hybrid molecules, five compounds, **125**, **128**, **129**, **134**, and **136** (Figure 16), have potential biological activities, and, according to docking studies, these compounds demonstrated strong hydrogen bonding interactions with the carboxylate group and the tetrazole moiety between Arg122A and Arg458D. According to the Swiss ADME server, compounds **134** and **136** displayed enhanced bioavailability, which was even better than the bioavailability of ciprofloxacin and norfloxacin. The results showed that the tetrazole moiety enhances pharmacological activity because of the high electron density of the nitrogen atoms in the tetrazole ring. The structural relationship between these two compounds, ciprofloxacin and norfloxacin, was analyzed, and it was shown that the biological activity is enhanced by the presence of groups that increase the molecular volume.

### 3.2. Anticancer

Cancer ranks as a leading cause of death on a global scale. According to the WHO, approximately 14 million new cancer cases and 9 million deaths related to cancer are reported each year. Alarmingly, it is also predicted that by 2030, the global number of cancer-related deaths will reach 13.1 million [84]. Ciprofloxacin has been shown in multiple publications to provide anticancer properties against some tumor cell lines, such as K562 (leukemia cells), U87MG (glioblastoma), MDA MB-231 (breast cancer), Colo829 (melanoma), Panc-1 (pancreatic cancer), H460 (lung cancer), LOVO (colon cancer), PC3, MLC9981 (prostate cancer), HT-29 (colorectal carcinoma), and HTB9 (bladder cancer) [85]. Oral administration of ciprofloxacin is a safe and efficient treatment option for many serious infections in cancer patients. Many of the current studies that are included in Table 2 indicate that ciprofloxacin has potent anticancer properties. Ciprofloxacin is reported to hinder mitochondrial topoisomerase II, leading to an impact on cellular energy metabolism. Ciprofloxacin triggers cell death in concentrations surpassing 80 mg/mL, whereas at a concentration of 25 mg/mL, it hinders the growth of Jurkat cells by preventing mitosis without causing any observable signs of cell death [86,87]. However, several studies have demonstrated that fluoroquinolones can stop cancer cells from proliferating through a variety of methods, such as DNA intercalation, apoptosis induction, and cell cycle arrest [88].

Given the features of natural fatty acids like biocompatibility, biodegradability, and enhanced cellular absorption by cancer cells, Chrzanowska et al. [29] hypothesized that linking them with ciprofloxacin will enhance their bioavailability and boost their cytotoxic potential. Ciprofloxacin was hybridized with either saturated or unsaturated fatty acids, as illustrated in Figure 17. Their impact on cytotoxicity, ability to induce apoptosis, and inhibition of IL-6 release across human primary (SW480) and metastatic (SW620) colon cancer, metastatic prostate cancer (PC3), and normal (HaCaT) cell lines were evaluated. The acyl groups introduced encompassed both unsaturated (**137**–**144**) and saturated (**145**) varieties, spanning short-chain (**137**, **138**), middle-chain (**139**), and long-chain (**140**–**145**) components. Certain members exhibited geometric *E*-isomerism (**137**–**139** and **141**), while others constituted naturally occurring *Z*-isomers (**140** and **145**–**147**). The synthesis occurred under mild conditions at room temperature, yielding good results. To ensure structural diversity, acids of varying chain lengths, degrees of unsaturation, and geometric isomerism were deliberately selected. The PC3 cell lines exhibited the highest sensitivity to the tested conjugates. Conjugate **140**, composed of oleic acid, demonstrated an IC_50_ value of 7.7 mM, significantly lower compared to ciprofloxacin alone (101.4 mM), indicating a 12-fold decrease in potency. These derivatives induced late apoptosis in all cancer cell lines with no notable effect on normal cells. Conjugate **140** emerged as the most potent apoptosis inducer, with 81.5% ± 3.9 of PC3 cells undergoing late apoptosis, followed closely by elaidic acid amide (**141**) at 75% ± 4.8. Among the studied derivatives, DHA (**144**) and sorbic (**138**) acid conjugates exhibited the strongest proapoptotic effects on SW480 cells, whereas compounds **138** and **141** showed remarkable efficacy in SW620 cell lines. To elucidate the cytotoxic mechanism of derivatives **128**, **140**, and **141**, interleukin-6 (IL-6) levels were measured, revealing a significant decrease in IL-6 release by cancer cells treated with the most cytotoxic conjugates. Additionally, all conjugates were assessed for in vitro antimicrobial activity, with crotonic (**137**) and sorbic (**138**) short-chain amides demonstrating potent activity against *Staphylococci*. Notably, sorbic acid amide (**138**) exhibited strong antimicrobial and antitumor properties, highlighting its dual functionality.

Ahadi et al. [88] modified the 3-carboxylic functionality of the ciprofloxacin structure to change its activity from antibacterial to anticancer activity. Therefore, a series of C-3-modified ciprofloxacin derivatives with an *N*-(5-(benzylthio)-1,3,4-thiadiazol-2-yl)-carboxamide moiety were successfully synthesized. Figure 18 illustrates a series of reactions employed for synthesizing innovative quinolone-based thiadiazoles. Initially, 5-amino-1,3,4-thiadiazole-2-thiol (**147**) was synthesized through the heterocyclization of thiosemicarbazide (**146**) using carbon disulfide and KOH in absolute ethanol. Compound **147** was then reacted with benzyl chloride or bromide derivatives to yield S-benzyl derivatives, **148**–**154**, in good yields. Conversely, to modify the 3-carboxylic acid functionality of ciprofloxacin, the NH of the piperazinyl group underwent protection with a tert-butyloxycarbonyl (Boc) group. This involved treating ciprofloxacin with Boc_2_O in the presence of sodium bicarbonate in THF-H_2_O, resulting in N-Boc-ciprofloxacin (**155**). To activate the 3-carboxylic acid, compound **155** was treated with ethyl chloroformate and TEA as a base in dry DCM to yield intermediate **156**. The reaction of compound **156** with aminothiadiazoles **148**–**154** produced the corresponding amides **157**–**163**. Finally, deprotection of the N-Boc-amides (**157-163**) with TFA in DCM at room temperature generated compounds **164**–**170**.

The majority of the compounds exhibited notable effects against MCF-7, A549, and SKOV-3 cancer cells in the MTT assay. Specifically, compounds **164**–**168** and **169** demonstrated potency comparable to the standard drug doxorubicin against the MCF-7 cell line (with IC_50_ values ranging from 3.26 to 3.90 µM). Additionally, the 4-fluorobenzyl derivative **170** displayed remarkable activity against SKOV-3 and A549 cells, with IC_50_ values of 3.58 and 2.79 µM, respectively, matching the potency of doxorubicin. Two notable compounds, **168** and **169**, were subjected to further assessment for their ability to induce apoptosis and arrest the cell cycle. Both compounds significantly prompted apoptosis in MCF-7 cells, with compound **168** being more potent, resulting in an 18-fold increase in apoptotic cell proportion at the IC_50_ concentration in MCF-7 cells. Cell cycle analysis demonstrated that compounds **168** and **170** increased cell populations in the sub-G1 phase, leading to oligonucleosomal DNA fragmentation and apoptosis, which was confirmed by the comet assay.

Kassab et al. [97] developed compounds with both anticancer and antibacterial properties. They utilized a biology-oriented drug synthesis approach to produce a range of new ciprofloxacin hybrid molecules. The reaction for synthesizing the desired compounds is described in Figure 19. The initial compound, **170**, was synthesized through the reaction between ciprofloxacin and ethyl chloroacetate in DMF. Compound **170** was reacted with hydrazine hydrate to yield compound **172**. The hydrazones **173**–**177** were synthesized by treating compound **172** with the respective aldehyde in ethanol. Some of the compounds were further investigated to determine their IC_50_ values against the most responsive cancer cell lines. In vitro experiments revealed that these five compounds displayed significant anticancer efficacy against the tested cell lines within the nanomolar to micromolar range, with IC_50_ values ranging from 0.72 to 4.92 mM. This potency ranged from 9 to 1.5 times greater than that of doxorubicin. Compounds **174** and **176** emerged as promising potent anticancer hybrids, demonstrating a strong correlation between their antiproliferative activity and their ability to inhibit topoisomerase II (with IC_50_ values of 0.58 and 0.86 mM, respectively). Compound **174** exhibited a 6-fold greater potency than doxorubicin, a 5-fold greater potency than amsacrine, and a 1.5-fold greater potency than etoposide. Similarly, compound **176** showed a four-fold stronger inhibition of topoisomerase II compared to doxorubicin, a three-fold greater potency than amsacrine, and nearly equivalent activity to etoposide. Activation of the DNA damage response pathway led to cell cycle arrest at the G2/M phase, accumulation of cells in the pre-G1 phase, and annexin-V and propidium iodide staining, indicating apoptosis as the mode of cell death. Furthermore, compounds **174** and **176** exhibited potent proapoptotic effects through induction of the intrinsic mitochondrial pathway of apoptosis, as evidenced by a significant increase in active caspase-3 levels compared to the controls. This suggests that both ciprofloxacin hybrids may chelate with zinc, a potent inhibitor of procaspase-3 enzymatic activity, thereby facilitating the processing of procaspase-3 into its active form. The newly synthesized ciprofloxacin derivatives were evaluated in vitro for their antibacterial effectiveness against *B. subtilis*, *S. aureus*, *E. coli*, and *P. aeruginosa* strains. The findings demonstrated that all tested compounds exhibited significant antibacterial properties, ranging from good to excellent, in comparison to ciprofloxacin. Notably, compound **177** displayed greater potency than ciprofloxacin, specifically against *P. aeruginosa*, a prevalent pathogen responsible for infections in patients with granulocytopenic cancer.

Swedan et al. [98] developed novel ciprofloxacin derivatives containing biologically active groups substituted at the C-7 position. Compound **178** was synthesized by a procedure outlined by Kassab et al. [97]. Compound **179** was synthesized by reacting compound **178** with diethyl malonate in the presence of sodium ethoxide (Figure 20). The hydrazones **180** and **181** were obtained through the reaction of compound **178** with the appropriate ketone in ethanol with glacial acetic acid. Compound **182** was obtained by refluxing compound **178** with the suitable isatin in absolute ethanol with glacial acetic acid. Compound **183** was prepared by reacting compound **178** with the appropriate phenyl isocyanates or phenyl isothiocyanate in absolute ethanol with glacial acetic acid. Ciprofloxacin derivative **184** was prepared by reacting compound **178** with phenacyl bromide in dry benzene with potassium carbonate.

The most potent compounds, **179** and **184**, exhibited significant inhibitory activity against Topoisomerase II, ranging from 83% to 90% at a concentration of 100 μM. Compounds **179**, **182**, and **184** demonstrated 1.01- to 2.32-fold higher potency compared to doxorubicin. Compounds **179** and **182** triggered apoptosis in T-24 cells, with respective increases of 16.8- and 20.1-fold compared to the control. This apoptotic effect was further confirmed by a substantial rise in apoptotic caspase-3 levels, increasing by 5.23- and 7.6-fold, respectively. Both compounds arrested the cell cycle at the S phase in T-24 cancer cells, while in PC-3 cancer cells, the cell cycle at the G1 phase was arrested. Molecular docking simulations of compounds **179** and **182** into the active site of topoisomerase II provided a rationale for their remarkable inhibitory activity against topoisomerase II.

Szostek and colleagues [70] evaluated all compounds synthesized in Figure 8 to determine their cytotoxic effects using the MTT method in vitro. Among the compounds tested, three compounds (**33**, **41**, and **46)** (Table 3, Figure 8) demonstrated significant activity against cancer cells while showing no cytotoxicity towards normal cells. The selectivity index (SI) of doxorubicin ranged from 0.14 to 1.11, whereas the SIs for the aforementioned three compounds ranged from 1.9 to 3.4. Certain derivatives of ciprofloxacin were subjected to molecular docking into the crystal structure of topoisomerase II (DNA gyrase) complexed with DNA (PDB ID: 5BTC). The promising lead structures were identified (i.e., **3-33**, **41**, **42**, and **46**). Previous studies on disubstituted derivatives yielded unsatisfactory results, indicating that the 4-oxo-3-carboxylic acid core serves as the active binding site for DNA gyrase. Structural modifications in this fragment resulted in a noticeable decrease in antibacterial efficacy. Compound **36** exhibited the highest SI of 3.4, demonstrating an IC_50_ value of 29.5 ± 2.1 µM against human colon cancer cells (HCT-116), while showing no cytotoxicity towards human immortal keratinocyte cells from adult human skin (HaCaT). The most significant cytotoxic effect was observed with compound **32**, with an IC_50_ of 16.5 ± 0.4 µM against human liver cancer cells (HepG2). Except for derivative **36**, the group of menthol derivatives showed cytotoxicity across all the cell lines, with most of the SI values being below 1.

Akhtar et al. [99] synthesized some novel *N*-acylated ciprofloxacin-based pharmacophores by reacting various sulfonyl halides and carboxylic acids, or acyl, with ciprofloxacin methyl ester. They started with commercially available ciprofloxacin and easily synthesized methyl ester **185** (70%) through Fischer esterification. *N*-Acylation of ciprofloxacin ester **185** using trifluoroacetic anhydride or acyl and sulfonyl halides in DCM with pyridine yielded the respective *N*-acylated ciprofloxacin analogues **186**, **187** (67% and 85%, respectively), **191**, and **192** (66% and 68%, respectively) [100,101,102]. Additionally, a range of carboxylic acids were reacted with DCC, employing ciprofloxacin ester **185**, resulting in targeted compounds **188**–**190**, with yields ranging from 77% to 87% (Figure 21). The findings indicated a decreased cell viability for compounds **186**, **187**, and **189** (42.66%, 50.68%, and 46.86%, respectively) at a concentration of 100 μg/mL. The half-maximal inhibitory concentration values (IC_50_) revealed the significance of **186**, demonstrating superior cytotoxicity compared to other synthesized derivatives with an IC_50_ value of 2.0 μg/mL. The presence of a good leaving group (Br) suggests that compound **186** holds promise for generating various *N* or *S*-alkylated ciprofloxacin derivatives with enhanced anticancer properties in future studies. The use of computational modeling (in silico) for compound **186** (with an IC_50_ of 2.0 μg/mL) provided additional clarity on its potential as an anticancer agent. Molecular docking studies showed that compound **186** exhibits a greater affinity for inhibiting topoisomerase II than ciprofloxacin.

Alaaeldin et al. [103] synthesized a novel C7-piperazinyl ciprofloxacin *ortho*-phenolic chalcone Mannich base derivative (**196**). In vitro anticancer evaluation of the compounds against HepG2 and A549 cancer cells was performed. Moreover, the mechanism of action of compound **196** was investigated to determine its capability to inhibit topoisomerase I/II activity and trigger apoptotic and necro-apoptotic pathways. Initially, the authors produced a chalcone using a technique outlined by Karki et al. [104], which involved mixing furfural and 4-hydroxy acetophenone in 15 mL of ethanol. Aqueous KOH solution was added to the mixture with vigorous stirring under an argon atmosphere at room temperature for 12–24 h. Following this, ethanol was evaporated under reduced pressure. The resulting residue was then diluted in ice water and acidified using a 10% hydrochloric acid solution. After filtration and water rinsing, the crude product was recrystallized with ethanol to yield the derivative **195.** An equimolar blend of ciprofloxacin and chalcone (**195**) was combined in ethanol, followed by the addition of 1 mL of formaldehyde (37%). The mixture was then heated under reflux for 36 h and subsequently cooled. The resulting solid was filtered under vacuum, and Mannich bases were obtained in a 58% yield through crystallization from aqueous ethanol (Figure 22).

The findings from molecular docking revealed that this new ciprofloxacin derivative, **196**, effectively binds to and inhibits both topoisomerase I and topoisomerase II. It demonstrated significant suppression of A549 and HepG2 cancer cell proliferation, along with reduced cell migration and colony formation capabilities. Moreover, it increased the percentage of apoptotic cells, induced cell cycle arrest at the G2/M phase, and elevated lactate dehydrogenase (LDH) release levels in both cancer cell lines. Additionally, it upregulated the expression of cleaved caspase 3, RIPK1, RIPK3, and MLKL proteins. This innovative ciprofloxacin derivative exhibited robust dual inhibition of topoisomerase I/II enzyme activities, displayed antiproliferative effects, inhibited cell migration and colony formation in A549 and HepG2 cancer cells, and triggered apoptotic pathways. Furthermore, it initiated an alternative cell death pathway, necroptosis, by activating the RIPK1/RIPK3/MLKL pathway.

Fathyn et al. [105] examined how a novel derivative of ciprofloxacin, specifically a 7-(4-(N-substituted carbamoylmethyl) piperazin-1-yl)-based compound, influenced the proliferation and migration capacities of HeLa cells. This compound, designated as compound **197**, was synthesized following a previously documented procedure [33]. This involved the alkylation of ciprofloxacin with acetylated chalcone derivatives in acetonitrile (Figure 23).

The investigated derivative of ciprofloxacin (**198**) demonstrated a decrease in the survival rate of HeLa cells in correlation with concentration and caused changes in the morphology of the cells, suggesting cell death. Moreover, it impeded the process of wound healing, even at concentrations that were not harmful to the cells and diminished the formation of HeLa cell colonies. Additionally, there was an increase in apoptosis, likely due to a significant rise in the expression of Bax protein and the activation of cleaved caspase-3 protein. In conclusion, their novel derivative impedes the growth and triggers apoptosis in HeLa cells, while also inhibiting their migration and colony formation abilities.

Shahzad et al. [106] developed ciprofloxacin analogues (Figure 24) and evaluated their ability to inhibit thymidine phosphorylase. The majority of these derivatives displayed thymidine phosphorylase inhibitory activity, with IC_50_ values ranging from 39.71 ± 1.13 to 161.89 ± 0.95 µM, when compared to the standard, 7-Deazaxanthine with an IC_50_ value of 37.82 ± 0.93 µM. Out of all the synthesized ciprofloxacin analogues, compound **200** showed the strongest inhibitory activity with an IC_50_ value of 39.71 ± 1.13 µM. Compound **200** was synthesized by esterification reaction via the carboxylic moiety of ciprofloxacin by treating ciprofloxacin with hydrochloric acid (37%) in the presence of dry methanol at room temperature. Dropwise additions of thionyl chloride (3.5 equiv) were made to the reaction mixture at 0 °C, followed by a further 24 h of reflux. Neutral conditions prevented piperazine moiety protonation or salt formation, resulting in the desired ester intermediate ciprofloxacin derivative **199**. Compound **200** was successfully synthesized by reacting the methyl ester of the intermediate ciprofloxacin derivative **199** with 2-phenyl acetyl chloride in the presence of anhydrous potassium carbonate under stirring at room temperature (Figure 24).

Fallica et al. [107] designed and synthesized twelve novel ciprofloxacin- and norfloxacin-based derivatives endowed with a nitric oxide photo-donor moiety to explore their potential synergistic antitumor effects.

The approach devised for producing compounds **201** and **202** is outlined in Figure 25a. Carboxylic acids **201** and **202** were synthesized through a single-step method aromatic nucleophilic substitution between 4-fluoro-1-nitro-2-(trifluoromethyl)benzene and ciprofloxacin in dimethyl sulfoxide (DMSO) 120 °C for 1 h. Figure 25b outlines the synthetic pathway employed to produce compounds **210**–**213** and **214**–**217**. Beginning with 4-fluoro-1-nitro-2-(trifluoromethyl)benzene, intermediates **206** and **207** were synthesized via an aromatic nucleophilic substitution with either 2-aminoethanol or 3-aminopropan-1-ol in acetonitrile at 60 °C overnight. Subsequently, mesylation of the alcohol groups in compounds **206** and **207** yielded intermediates **208** and **209**, which were then subjected to reaction with derivative **203** under reflux with acetonitrile overnight. The resulting methyl esters, **210**–**213**, were hydrolyzed via reflux with a 2 M NaOH aqueous solution for 24 h to yield the respective carboxylic acids, **214**–**217**. All the synthesized fluoroquinolone derivatives demonstrated potent anticancer activity against a panel of various cancer cell lines, and compounds **214**–**217** were the most effective compounds. The results revealed that some of the tested compounds, such as **201**, **202**, and **214**–**217**, displayed potent cytotoxic effects on MDA-MB231 tumor cells. Furthermore, **201** and **202** induced a potent cytotoxic response in DOX-resistant MCF7/ADR breast cancer cells.

Ezelarab et al. [108] developed Ciprofoxacin-piperazine C-7 linked quinoline derivatives and investigated their antiproliferative, antibacterial, and antifungal activities. The synthetic strategy developed for producing the targeted compounds is outlined in Figure 26. The initial suitable acids, **218**–**220**; the intermediate esters, **221**–**223**; and hydrazides **224**–**226** were synthesized using the established methods outlined in the existing literature [109]. Hydrazides **224**–**226** were transformed into 1,3,4-oxadiazole-2-thione derivatives **227**–**229** via a previously described method [110] involving refluxing in an ethanolic solution with CS_2_ alongside KOH serving as a potent base. The reaction for synthesizing acylated ciprofloxacin **233** followed the documented method [111] involving treating ciprofloxacin with bromoacetyl bromide in an ice bath alongside potassium carbonate as a base. The alkylation of oxadiazole derivatives **227**–**229** with the resulting acylated ciprofloxacin **233** was conducted in acetonitrile with TEA, resulting in the production of new hybrids, **234**–**236**, at satisfactory levels.

Compounds **234** and **235** demonstrated antiproliferative effects against SR-leukemia cell lines, with growth inhibition rates of 33.25% and 52.62%, respectively. These findings suggest that electron-donating substitutions, such as a p-methyl group on the phenyl ring, as seen in compound **235**, are more advantageous than having an unsubstituted phenyl ring at this position, as observed in compound **234**. Conversely, replacing the p-methyl group with a p-methoxy group resulted in a loss of antiproliferative activity, as evidenced by compound **236**. This underscores the significant impact of these compounds’ structures on their anticancer potential.

In terms of CAKI-1 renal cancer, compound **235** demonstrated enhanced anticancer efficacy compared to hybrids **234** and **236**, with growth inhibition rates of 39.81%, 26.92%, and 27.31%, respectively. Additionally, hybrids **234**–**236** displayed notable antiproliferative effects against UO-31 renal cancer, with growth inhibition rates of 64.19%, 55.49%, and 40.15%, respectively.

Additionally, concerning LOX IMVI melanoma cancer cells, compounds **234** and **235** demonstrated significant anticancer effects, inhibiting cell growth by 39.14% and 36.64%, respectively. Conversely, compound **236** exhibited minimal anticancer activity, with only 8.95% growth inhibition. This suggests that the incorporation of a stronger electron-donating group, such as the methoxy group, into the phenyl ring at position 2 of the quinoline ring in compound **236** had a detrimental impact on its anticancer efficacy compared to compounds **234** and **235**. Thus, having an unsubstituted phenyl ring at this position, as in compound **234**, proved more effective in inhibiting the proliferation of LOX IMVI melanoma cells. Similar trends were observed in the growth inhibition rates of compounds **234**–**236** for A498 renal cancer cell lines.

Mohammed et al. [81] synthesized a novel class of ciprofloxacin–chalcone-based hybrid molecules and examined their potential cytotoxicity against NCI-60 cancer cell lines. The synthesis of the new hybrids **240**–**241** and their corresponding intermediate, compound **239**, was carried out according to the reactions outlined in Figure 27. The carbamate ester **238** was prepared by stirring *p*-amino-acetophenone (**237**) with ethyl chloroformate in acetonitrile, using pyridine as a base. The preparation of ciprofloxacin derivative **239** was achieved by refluxing ciprofloxacin with the ester intermediate **238** in xylene. Condensation of the main intermediate, compound **239**, with various aromatic aldehydes in an ethanolic sodium hydroxide solution furnished the target hybrids **240**–**241**.

Among these hybrids, compounds **240** and **241** demonstrated exceptional antiproliferative activities against leukemia SR and colon HCT-116 tumor cell lines in comparison to camptothecin, topotecan, and staurosporine. Additionally, hybrids **240** and **241** showed inhibition of topoisomerase I. Compound **241** inhibited the leukemia SR cell lines’ cell cycle at the G2/M phase. It induced apoptosis both extrinsically and intrinsically by activating the proteolytic caspase cascade, releasing cytochrome C from mitochondria, upregulating proapoptotic Bax, and downregulating Bcl-2 protein levels. The new hybrids **240** and **241**, as well as their corresponding intermediate compound **239**, were synthesized using the reactions shown in Figure 27. The derivative **239** was synthesized by refluxing ciprofloxacin and the ester intermediate **238** in xylene. The target hybrid compounds **240** and **241** were obtained by condensing the main intermediate, compound **239**, with different aromatic aldehydes in an ethanolic NaOH solution.

Samir et al. [112] reported the design and synthesis of a novel series of ciprofloxacin 3,7-bis-benzylidenes hybrid compounds. The compounds **244**–**246** were synthesized following the synthetic pathway described in Figure 28. The diester fluoroquinolone **242** was synthesized through a reaction involving ciprofloxacin and two equivalents of ethyl chloroacetate in DMF, with K_2_CO_3_ acting as a catalyst. Hydrazinolysis of the diester **242** with hydrazine hydrate afforded ciprofloxacin bis-hydrazide **243.** Consequently, the acylhydrazone derivatives **244**–**246** were prepared by the reaction of **243** with the appropriate aldehyde in ethanol and glacial acetic acid as catalysts.

Most of the target compounds displayed effective cytotoxicity, with the most potent variants, **244** and **246** demonstrating significant broad-spectrum antiproliferative effects comparable to Doxorubicin against HL-60 (TB), leukemia, HCT-116 colon cancer, and MCF7 breast cancer cell lines. Additionally, derivative **246** induced apoptosis at the G2/M phase. Evaluation of the mechanism of action of compounds **244**–**246** revealed promising dual inhibition of topoisomerase Iα (TOP Iα) and topoisomerase IIB (TOP IIB), akin to Camptothecin and Etoposide, respectively. Docking studies of **244**–**246** into the active sites of topoisomerase I and II proteins, compared to Camptothecin and Etoposide, indicated favorable binding scores and enhanced enzyme assay results. Therefore, compounds **245** and **246** show potential as targeted antiproliferative agents with dual action against TOP Iα and TOP IB.

### 3.3. Antiviral

While ciprofloxacin is not typically used as an antiviral agent, there has been some research exploring its potential antiviral properties, particularly against certain viruses. Some in vitro studies have suggested that ciprofloxacin and other fluoroquinolones may have antiviral activity against certain viruses. For example, ciprofloxacin has been investigated for its activity against viruses like BK virus [113], SARS-CoV-2/MERS-CoV [25], and human HIV/HCV viruses [114]. The potential antiviral activity of ciprofloxacin may be attributed to its ability to interfere with viral replication processes, similar to its action on bacterial DNA gyrase [115]. However, the exact mechanisms by which ciprofloxacin inhibits viral replication are not fully understood and may vary depending on the virus. While some promising results have been reported in laboratory studies, there is limited clinical evidence supporting the use of ciprofloxacin as an antiviral agent. Clinical trials evaluating its efficacy, specifically against viral infections, are lacking. Many scientists worldwide have been motivated by the antiviral qualities of ciprofloxacin to synthesize and alter its structure to develop potent molecules. RNA viruses like hepatitis C, zika, and dengue viruses have been previously reported to be suppressed by fluoroquinolones and their derivatives [116,117,118,119]. Ciprofloxacin displays antiviral efficacy against the BK virus by preventing its proliferation in an in vitro culture using the Vero cell lines [113]. In a study by Marciniec et al. [26], it was reported that moxifloxacin and ciprofloxacin have a considerable potential to bind to COVID-19 Main Protease (M^pro^). The GOLD docking studies revealed that moxifloxacin and ciprofloxacin bind to the protein active site more firmly than the natural ligand. A thorough examination of the ligand–protein interactions reveals that, when compared to the positive controls, the tested fluoroquinolones exert more protein interactions than chloroquine and nelfinavir.

Alavala et al. [120] performed a microwave-assisted synthesis of a wide range of dihydropyrimidine *N*- and O-Mannich bases of ciprofloxacin derivatives. Chalcones, denoted as α, β-unsaturated ketones in Figure 29 (**248**), were synthesized via crossed aldol condensation. This process involved the combination of aromatic methyl ketones with various aromatic aldehydes, each featuring electron-releasing or electron-withdrawing substituents positioned at meta or para positions. Dihydropyrimidinones (**259**, as shown in Figure 29) were produced through the reaction of the corresponding chalcone with urea and potassium hydroxide in ethanol. The resultant compounds were purified via recrystallization. Regioselective alteration of dihydropyrimidinones was accomplished through the use of a mild base, such as potassium carbonate. Mannich bases, derived from dihydropteridine reductase compounds **250**–**252** and **253**, were produced by reacting them with secondary amines and formaldehyde, with or without the presence of a small amount of potassium carbonate. This process was conducted using both traditional and microwave-assisted methods, as detailed in previous studies [121,122,123,124]. Utilizing industry-standard protocols, all the synthesized compounds were assessed for their cytotoxicity and antiviral activity against viral strains of the flu (A and B), hepatitis (B and C), West Nile, dengue severe acute respiratory syndrome, vaccinia, cowpox, and Venezuelan equine encephalitis viruses. Molecules **250**, **251**, **252**, and **253** were effective against hepatitis B, yellow fever, vaccinia, and the Tacaribe virus (TCRV), respectively. O-Mannich base ciprofloxacin molecules were shown to be more effective than *N*-Mannich base molecules, which also had a higher selectivity index.

### 3.4. Antimalarial Activity

Malaria is a protozoan parasitic infection caused by five parasite species belonging to the *Plasmodium* genus and is transmitted exclusively through the bite of female Anopheles mosquitoes. There are four types of human malaria species, namely, *Plasmodium falciparum* (*P. falciparum*), *Plasmodium vivax* (*P. vivax*), *Plasmodium ovale* (*P. ovale*), and *Plasmodium malariae* (*P. malariae*). They are responsible for the spread of malaria from person to person, which is facilitated by the Anopheles mosquito vector when taking a blood meal. *P. falciparum*, however, has long been recognized as the most lethal of the parasite species due to its high mortality rates and significant prevalence within sub-Saharan Africa [125]. Dana and colleagues [35] developed a novel category of hybrid compounds derived from ciprofloxacin, incorporating active antimalarial scaffolds such as acridine, quinolone, sulphonamide, and cinnamoyl pharmacophores. Among these hybrids, a potent antimalarial compound was **261**, which contains chloroquinolone and ciprofloxacin pharmacophores.

The synthesis of compound **261** commenced with the ethyl ester of ciprofloxacin (Figure 30). Compound **260** was derived using a previously reported method [126]. The key quinolone (**259**) was obtained by reacting 2,4,5-trifluorobenzoic acid with Vilsmeier–Haack reagent in DCM to yield acyl chloride (**256**) in good yield. Subsequently, this was condensed with ethyl 3-(diethylamino)acrylate (**254**) in a mixture of TEA/toluene, yielding *N,N*-diethyl enaminone intermediate (**257**) in a 76% yield. The synthesis of ethyl 3-(diethylamino)acrylate (**254**) involved a Michael addition of diethylamine to ethyl propionate in acetonitrile with an 88% yield, as previously reported. In a single-step process, transaminolysis of **257** with cyclopropylamine followed by cyclization with potassium carbonate in DMF produced difluoroquinolone (**259**) in 68% yield. The ethyl ester of ciprofloxacin (**260**) was synthesized through an aromatic nucleophilic substitution of the C-7 fluorine atom of difluoroquinolone **259** with piperazine, with a 98% yield. Finally, a nucleophilic substitution reaction between the secondary amine of the piperazine moiety at the C-7 position of **260** and the corresponding chloro-substituted first life cycle active antimalarial scaffolds (XCl) yielded the final ciprofloxacin-based hybrid compound, **261**, with yields ranging from 30% to 40%. Importantly, this compound exhibits a mechanism of action distinct from its parent molecules, revealing a novel biological target. Compound **261** demonstrated a significant effect against chloroquine-resistant and -susceptible strains of *P. falciparum* at low nanomolar concentrations (with IC_50_ values of 63.17 ± 1.2 nM and 25.52 ± 4.45 nM, respectively). Furthermore, it did not induce toxic effects on mammalian or bacterial systems at concentrations up to 20 μM and 1 μM, respectively.

A hybrid compound (**163**) combining dihydroartemisinin (DHA) and ciprofloxacin was synthesized by Ajima et al. [127] and tested for its effectiveness against the 3D7 strain of *P. falciparum*. The rationale behind this synthesis was to merge the artemisinin component, known for its action on the heme detoxification pathway of the malaria parasite, with the fluoroquinolone scaffold found in ciprofloxacin, which targets the plasmodial DNA gyrase enzyme. The hybrid was synthesized through an esterification reaction (Figure 31), combining ciprofloxacin’s carboxyl group with dihydroartemisinin’s hydroxyl group. It exhibited strong antimalarial effects against the tested *P. falciparum* strain, showing 3- to 4-fold greater potency (IC_50_: 2.99 nM) compared to chloroquine (IC_50_: 13.003 nM) and DHA alone (IC_50_: 9.968 nM). Furthermore, its in vitro cytotoxicity assessment revealed relatively low cytotoxicity (LC_50_: 50.78 µg/mL) when compared to cyclophosphamide (LC_50_: 1.08 µg/mL). In silico analysis of the hybrid’s Lipinski properties indicated favorable drug-like characteristics. Given its strong activity and low toxicity, this hybrid is a promising candidate for further development of new antimalarial drugs.

Vamvoukaki and colleagues [128] detailed the synthesis and testing of novel compounds merging the key structures of artemisinin and ciprofloxacin, along with 7-chloroquinoline, for their antiplasmodial properties (Figure 32). The initial step in all syntheses involves obtaining crucial piperazine ester intermediates containing ciprofloxacin. Using these platforms, a total of 18 final compounds were produced through a multistep reaction, with overall yields between 8% and 20%. Each compound was screened against the chloroquine-resistant *P. falciparum* FcB1 strain. Compounds **264**, **270**, **271**, and **272**, containing artesunate and ciprofloxacin pharmacophores, demonstrated IC_50_ values ranging from 3.5 to 5.4 nM, along with impressive selectivity indices. The most potent compounds were further tested against the ciprofloxacin-resistant Dd2 strain of *P. falciparum*, revealing that those incorporated with the artesunate moiety were effective. Combining artesunate with either ciprofloxacin or norfloxacin in a single molecular entity significantly enhanced the activity and selectivity compared to administering the unconjugated counterparts of artesunate/ciprofloxacin and artesunate/norfloxacin.

## 4. Conclusions and Future Directions

Ciprofloxacin is a fluoroquinolone molecule that is associated with different biological activities. This review article emphasizes the diverse biological activities of ciprofloxacin derivatives and outlines the different methods for developing ciprofloxacin derivatives. In recent years, extensive research has been conducted to advance ciprofloxacin scaffold development. The unique biological characteristics of these novel ciprofloxacin derivatives suggest promising potential for advancing the ciprofloxacin pharmacophore as a precursor for the development of safer and more efficient therapeutic agents in the future. We have meticulously outlined the comprehensive synthetic methodologies for the most potent derivatives in each study, yielding a total of 32 schemes for 150 derivatives, along with their potential biological effects. Modifying the 4-oxo-3-carboxylic acid functionality of ciprofloxacin led to a reduction in its effectiveness against bacterial strains. Thus, most synthetic strategies avoided modifying the 4-oxo-3-carboxylic moiety for antibacterial activity. Combining ciprofloxacin with other bioactive scaffolds through hybridization represents a prevalent and promising approach for enhancing its biological effects. This method has been used in most of the reported studies aimed at developing novel compounds derived from ciprofloxacin, showing encouraging biological activities.

One of the biggest challenges with ciprofloxacin and its derivatives is the emergence of drug-resistant microbes. Resistance can occur through mutations in the bacterial DNA gyrase and topoisomerase IV genes, as well as through efflux pumps and reduced membrane permeability. Developing ciprofloxacin derivatives that can effectively combat resistant strains without contributing to further resistance is a key concern. While the goal of modifying ciprofloxacin is to improve its safety and efficacy, modifications to the chemical structure can result in unexpected toxicities or side effects. Fluoroquinolones are known for their potential to cause severe adverse effects, such as tendonitis, tendon rupture, central nervous system disturbances, and phototoxicity. Developing derivatives with a better safety profile is a significant challenge.

Even if ciprofloxacin derivative shows promise in preclinical studies, moving to clinical trials and regulatory approval is a lengthy and expensive process. To overcome these challenges, performing in silico studies using computational tools, such as CARDIAC TOXICITY, PROTOX-II, SWISS ADME, etc., can be highly beneficial for predicting various properties and activities of chemical compounds. In silico approaches also accelerate the discovery of lead compounds with favourable drug-like properties.

This review article offers valuable insights for medicinal chemists seeking to explore future synthetic strategies for enhancing the efficacy and safety of ciprofloxacin-based compounds for addressing various diseases and disorders.

## Figures and Tables

**Figure 1 ijms-25-04919-f001:**
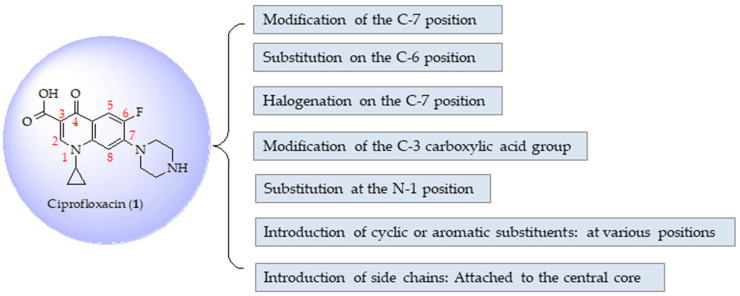
The chemical structure of ciprofloxacin with possible modifications.

**Figure 2 ijms-25-04919-f002:**
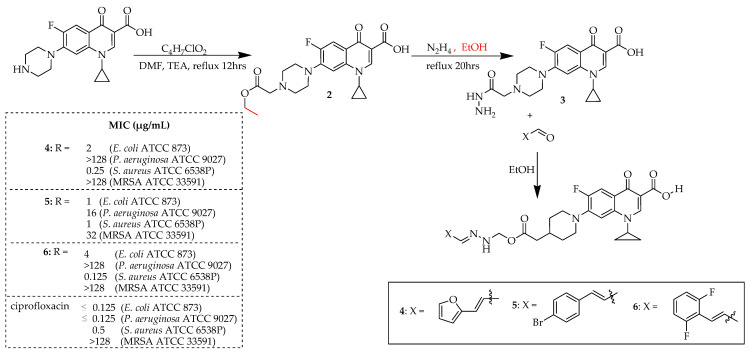
Synthetic strategy and antibacterial outcomes of *N*-acylarylhydrazone-ciprofloxacin derivatives (**4**–**6**).

**Figure 3 ijms-25-04919-f003:**
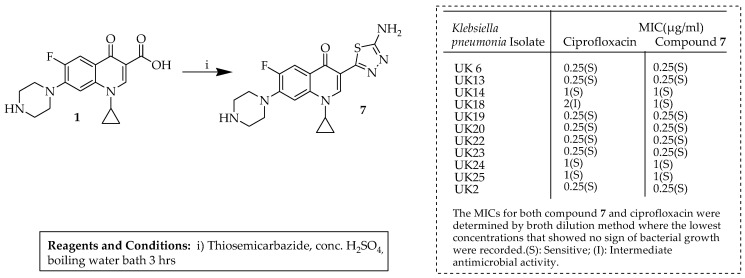
Synthetic strategy and the antibacterial outcomes of 1,3,4-thiadiazole–ciprofloxacin hybrid (**7**).

**Figure 4 ijms-25-04919-f004:**
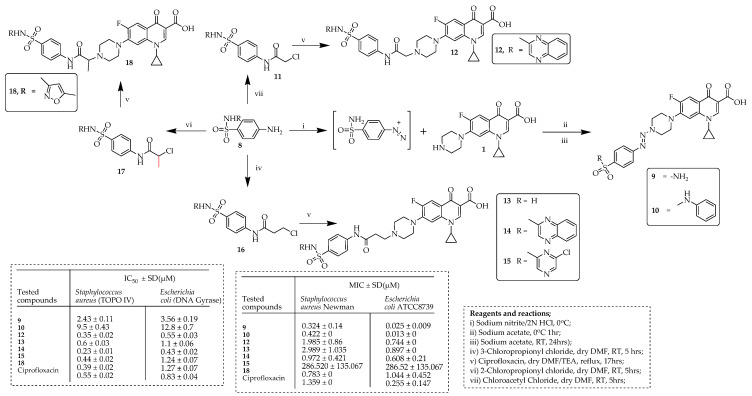
Synthetic strategy and the antibacterial outcomes of ciprofloxacin–sulfonamide hybrids (**9**–**18**).

**Figure 5 ijms-25-04919-f005:**
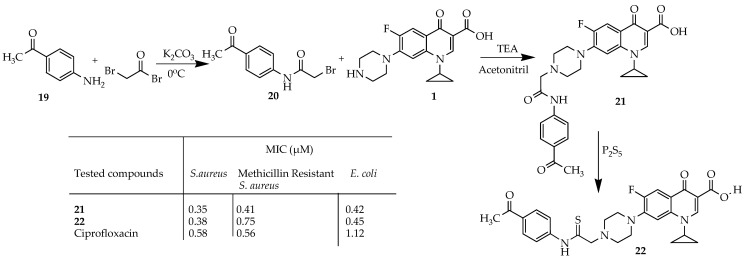
Synthetic strategy and the antibacterial outcomes of ciprofloxacin–sulfonamide hybrids (**21** and **22**).

**Figure 6 ijms-25-04919-f006:**
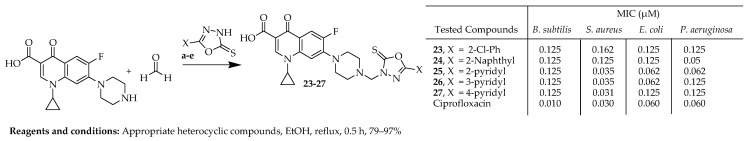
Synthetic strategy and antibacterial outcomes of oxadiazole derivatives (**23**–**27**).

**Figure 7 ijms-25-04919-f007:**
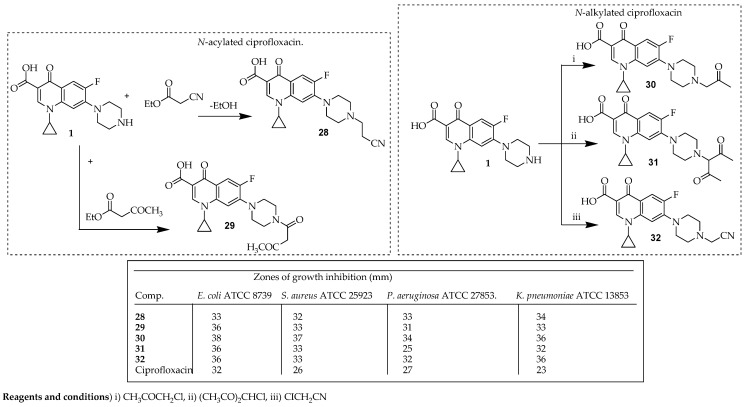
Synthetic strategy and the antibacterial outcomes of ciprofloxacin–cyanacetylpiprazinyl dihydroquinoline (**28**), –oxobutanoylpiprazinyl dihydroquinoline (**29**), and *N*-alkylated derivatives (**30**–**32**).

**Figure 8 ijms-25-04919-f008:**
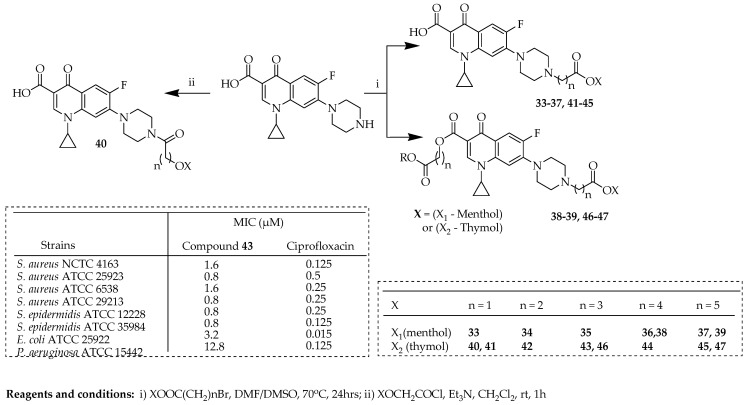
Synthetic strategy and the antibacterial outcomes of ciprofloxacin hybrids (**33**–**47**).

**Figure 9 ijms-25-04919-f009:**
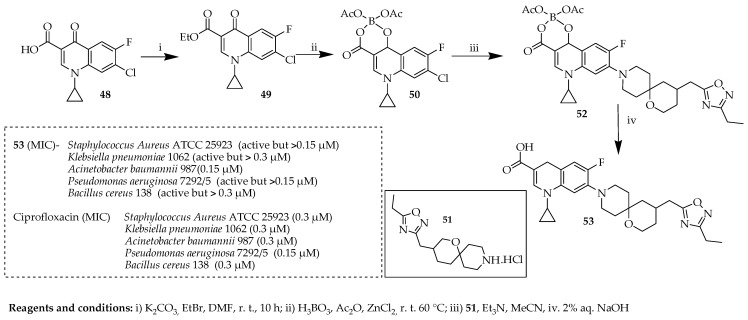
Synthetic strategy and the antibacterial outcomes of spirocyclic-periphery fluoroquinolones (**53**).

**Figure 10 ijms-25-04919-f010:**
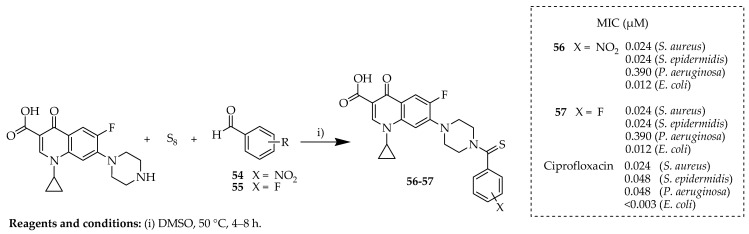
Synthetic strategy and the antibacterial outcomes of *N*-thioacylated ciprofloxacin derivatives **56**–**57.**

**Figure 11 ijms-25-04919-f011:**
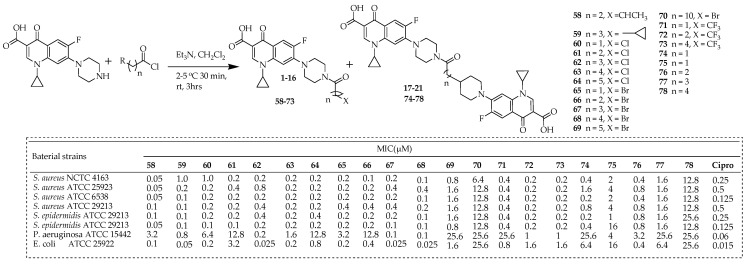
Synthetic strategy and the antibacterial outcomes of ciprofloxacin derivatives (**58**–**78**).

**Figure 12 ijms-25-04919-f012:**
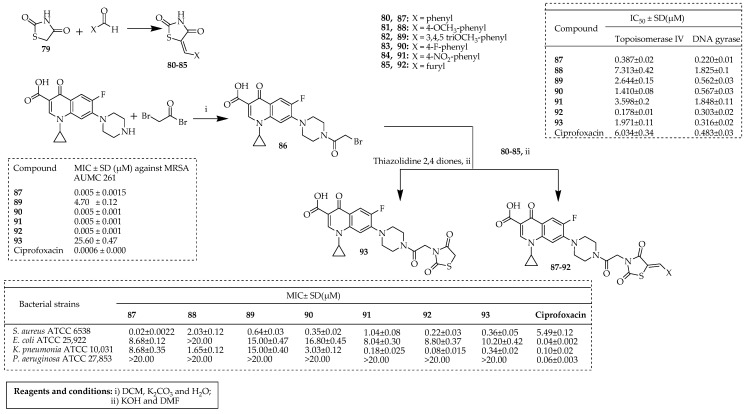
Synthetic strategy and the antibacterial outcomes of derivatives (**87**–**93**).

**Figure 13 ijms-25-04919-f013:**
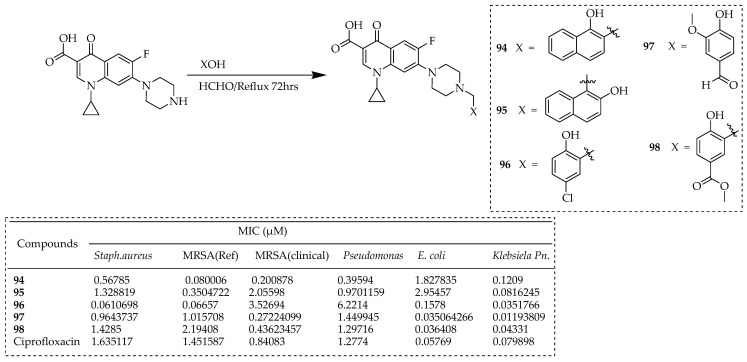
Synthetic strategy and the antibacterial outcomes of Mannich-based derivatives (**94**–**98**).

**Figure 14 ijms-25-04919-f014:**
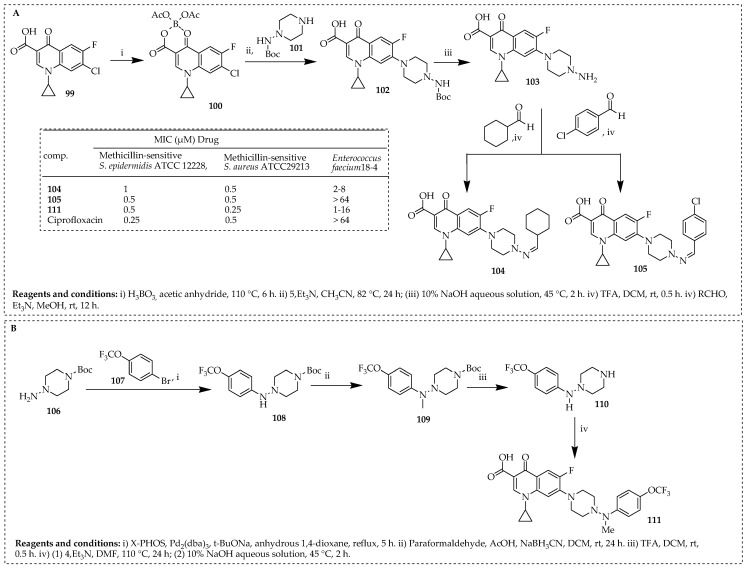
(**A**,**B**). Synthetic strategy and the antibacterial outcomes of derivatives (**104**, **105**, and **111**).

**Figure 15 ijms-25-04919-f015:**
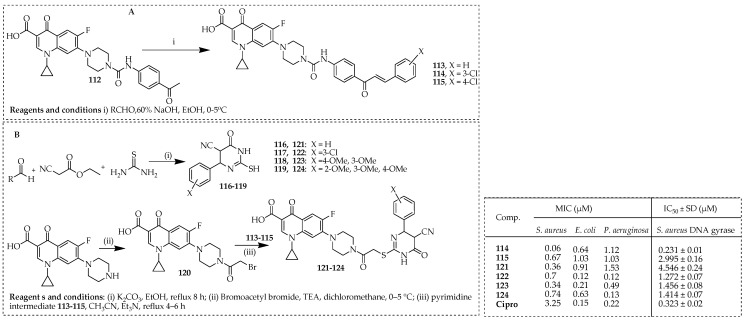
Synthetic strategy and the antibacterial outcomes of ciprofloxacin derivatives (**113**–**115** (**A**) and **121**–**124** (**B**)).

**Figure 16 ijms-25-04919-f016:**
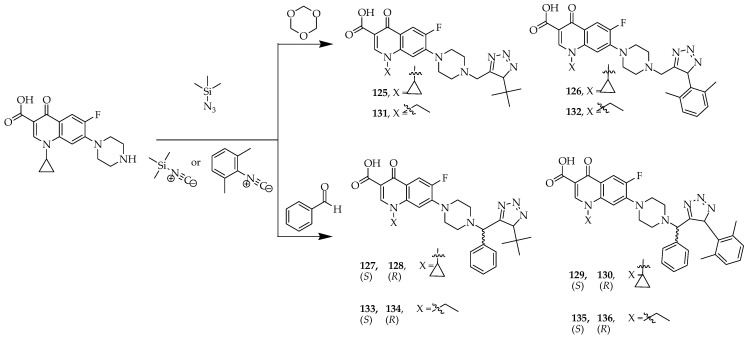
Synthetic strategy and the antibacterial outcomes of ciprofloxacin–sulfonamide hybrids (**125**–**136**).

**Figure 17 ijms-25-04919-f017:**
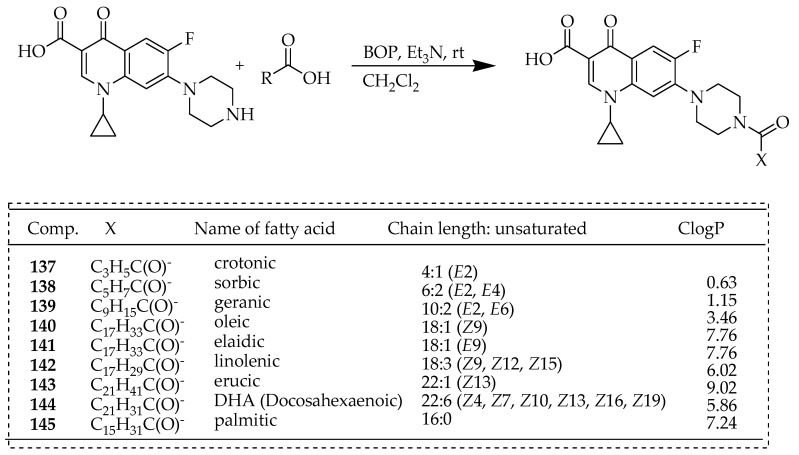
Synthetic strategy and anticancer outcomes of derivatives (**137**–**145**).

**Figure 18 ijms-25-04919-f018:**
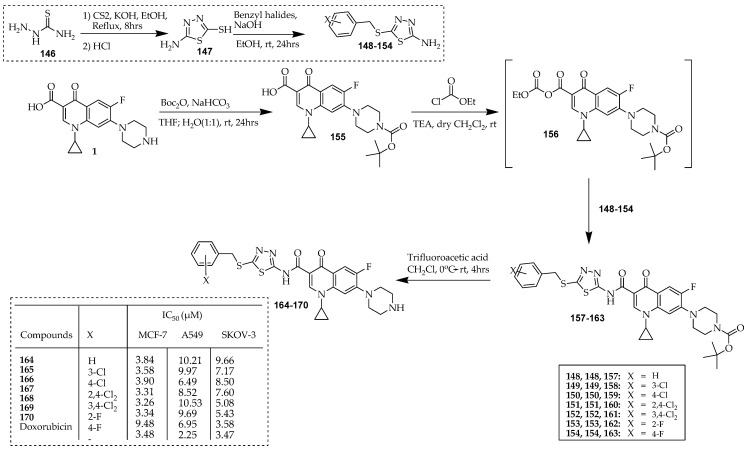
Synthetic strategy and anticancer outcomes of derivatives (**164**–**170**).

**Figure 19 ijms-25-04919-f019:**
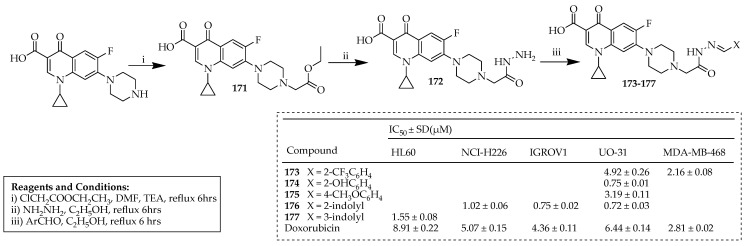
Synthetic strategy and the anticancer outcomes of ciprofloxacin–sulfonamide hybrids (**173**–**177**).

**Figure 20 ijms-25-04919-f020:**
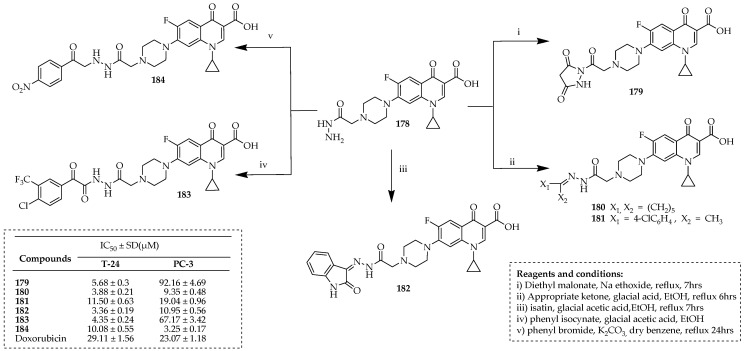
Synthetic strategy and the anticancer outcomes of derivatives (**179**–**184**).

**Figure 21 ijms-25-04919-f021:**
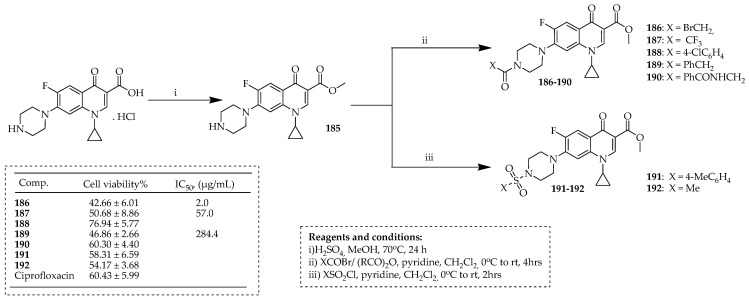
Synthetic strategy and the anticancer outcomes of *N*-acylated ciprofloxacin-based derivatives (**186**–**192**).

**Figure 22 ijms-25-04919-f022:**
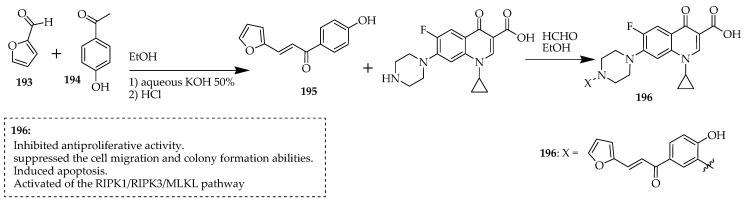
Synthetic strategy and the anticancer efficacy of C7-piperazinyl ciprofloxacin *ortho*-phenolic chalcone Mannich base derivative (**196**).

**Figure 23 ijms-25-04919-f023:**
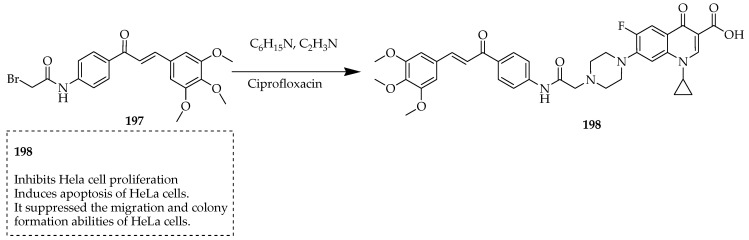
Synthetic strategy and the anticancer outcomes of derivatives **197**–**198.**

**Figure 24 ijms-25-04919-f024:**
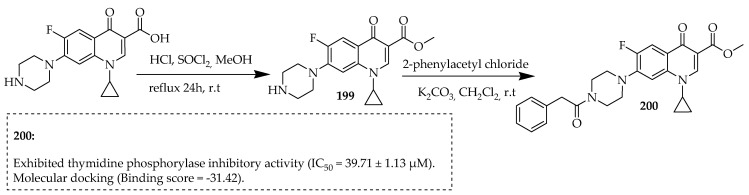
Synthetic strategy and the anticancer outcome of methyl 1-cyclopropyl-6-fluoro-1,4-dihydro-4-oxo-7-(4-(2-phenylacetyl)piperazin-1-yl)quinoline-3-carboxylate (**200**).

**Figure 25 ijms-25-04919-f025:**
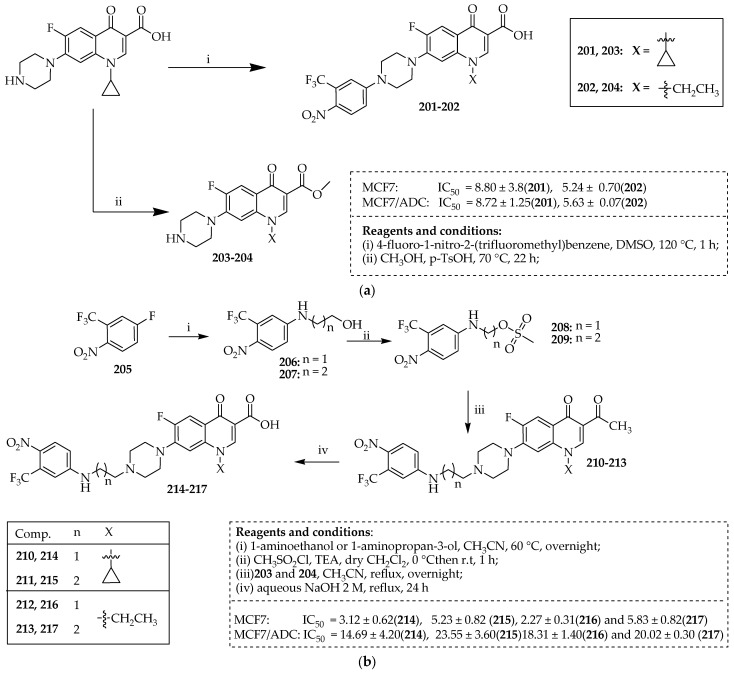
(**a**) Synthetic strategy and the anticancer outcomes of derivatives **201**–**202**. (**b**) Synthetic strategy and the anticancer outcomes of derivatives **214**–**217.**

**Figure 26 ijms-25-04919-f026:**
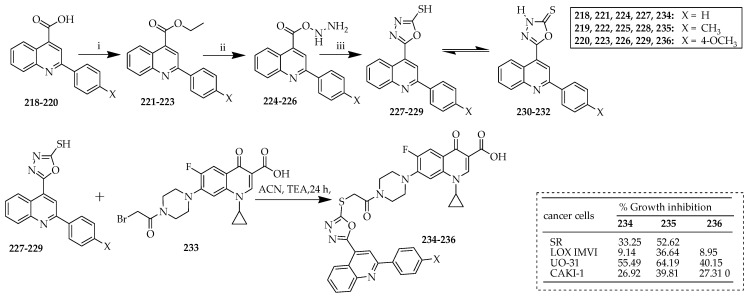
Synthetic strategy and the anticancer outcomes of ciprofloxacin–piperazine C-7 linked quinoline derivatives **234**–**236**.

**Figure 27 ijms-25-04919-f027:**
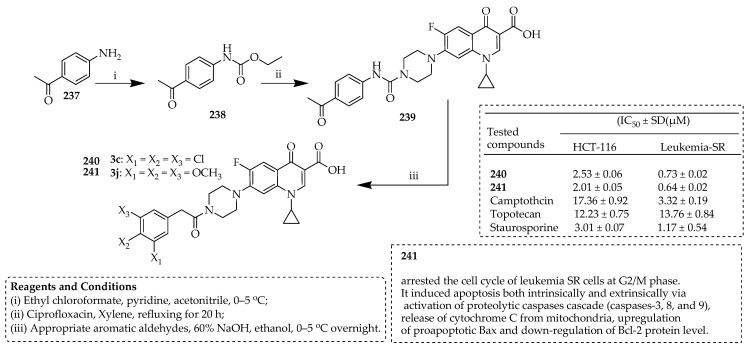
Synthetic strategy and the anticancer outcomes of ciprofloxacin–chalcone-based hybrids (**240**–**241**).

**Figure 28 ijms-25-04919-f028:**
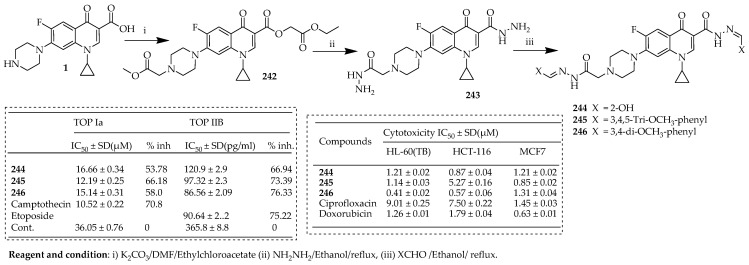
Synthetic strategy and the anticancer outcomes of ciprofloxacin–3,7-bis-benzylidenes hybrid compounds (**244**–**246**).

**Figure 29 ijms-25-04919-f029:**
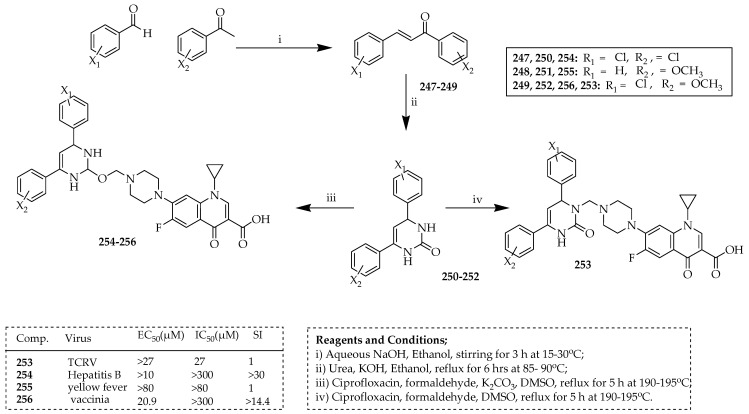
Synthetic strategy and the antiviral outcomes of dihydropyrimidine *N*- and O-Mannich bases of ciprofloxacin derivatives (**253**–**256**).

**Figure 30 ijms-25-04919-f030:**
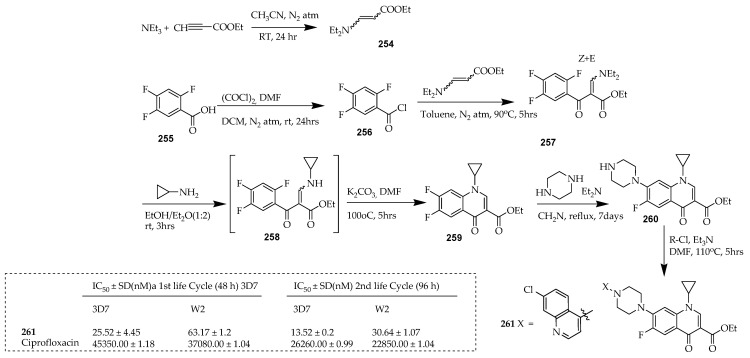
Synthetic strategy and the antimalarial outcomes of derivative **261.**

**Figure 31 ijms-25-04919-f031:**
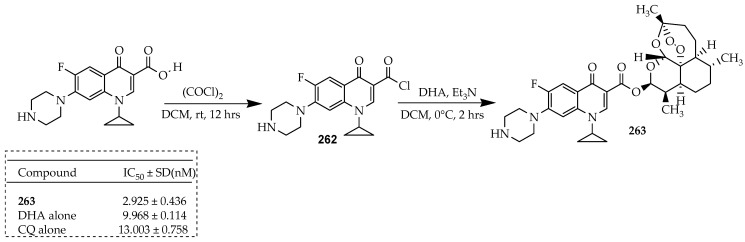
Synthetic strategy and the antimalarial outcomes of dihydroartemisinin-based ciprofloxacin derivative (**263**).

**Figure 32 ijms-25-04919-f032:**
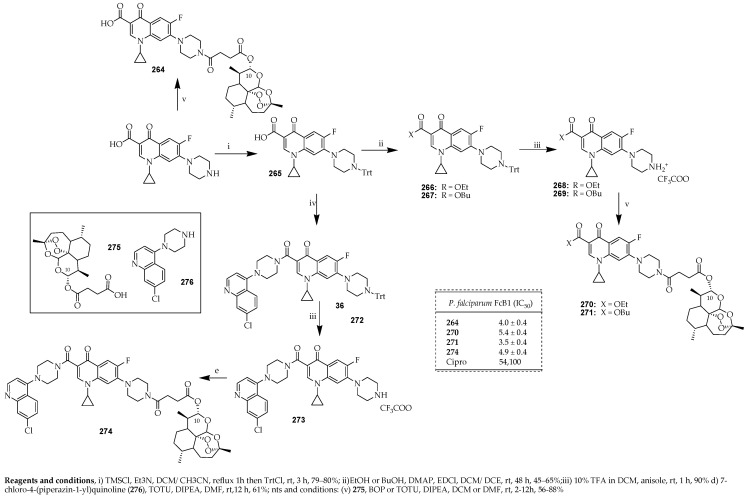
Synthetic strategy and the antimalarial outcomes of ciprofloxacin-based artemisinin hybrids (**264**–**274**).

**Table 2 ijms-25-04919-t002:** The potential of ciprofloxacin to combat cancer, along with its mechanism of action, decreased (↓) and increased (↑).

Cancer Cells	Dose and Time	Mode of Anticancer Activity	Mechanism of Action	Bibliography
Leukemia (K562)	>50 μg/mL	Inhibits proliferation and colony formation		[89]
Breast (MDA MB-231)	0–1.0 μmol/mL (0–72 h)	Apoptosis, S-phase arrest	Oxidative stress, p53 (↑), Bax:Bcl2 (↓)	[90]
Bladder cancer (HTB9)	50–400 μg/mL	Apoptosis, S-/G2-phase arrest	Cytochrome c release (↑), Bax:Bcl2 (↑), p21 (↓), Caspase3 (↑), pCDK2 (↓), Cyclin B/E (↓), CDK2 (↑)	[91]
CC-531, SW-403, HT-29 (colorectal carcinoma)	200–500 μg/mL (24 h)	Apoptosis	Bax:Bcl2 (↑), Caspase-3, -8, -9 (↑)	[92]
TK6, WTK1, NH32 (lymphoblastoid cells)	100 μg/mL (24–48 h)	G2-phase arrest and apoptosis	γH2AX (↑), Csp3 (↑), stabilized TopoIIα (↑)	[93]
HT29, Caco-2 (colon cancer)	100 μg/mL (6 days)	S-phase arrest	TGFβ1 (↑)	[94]
HeLa, A431 (epidermoid carcinoma)	3.5 W/cm^2^ UVA (30 min) + 100 μg/mL (24 h)	Apoptosis, S-/G2-phase arrest	DNA plasmid photocleavage via carbocation	[95]
H460 (lung cancer)	10 μg/mL (0–7 days)	Cancer stem cells	CD133 (↑), CD44 (↑), ABCG2 (↑), Oct4 (↑), Sox2 (↓), AKT (↑), ALH1A1 (↑), Slug (↓), Nanog (↑), Snail (↓), Cav-1 (↑), ERK (↑)	[96]

**Table 3 ijms-25-04919-t003:** The cytotoxic potential of the promising compounds as assessed through the MTT assay by Szostek et al. [70].

	Cancer Cells								Normal Cells
	HepG2		SW620		SW480		HCT116		HaCaT
Compounds	SI	IC_50_	SI	IC_50_	SI	IC_50_	SI	IC_50_	IC_50_
**32**	1.5	16.5 ± 0.4	0.9	26.7 ± 0.2	0.7	36.0 ± 2.9	0.8	32.3 ± 1.5	25.7 ± 3.0
**33**	1.2	36.8 ± 3.8	1.2	38.6 ± 3.8	1.5	30.3 ± 1.2	1.7	27.1 ± 3.1	45.5 ± 5.1
**36**	1.9	53.7 ± 0.9	2.6	38.1 ± 2.5	2.2	46.2 ± 1.8	3.4	29.5 ± 2.1	>100
**41**	1.9	51.3 ± 4.1	2.2	43.5 ± 7.5	2.9	33.7 ± 6.6	2.5	39.1 ± 5.8	>100
**42**	1.4	43.4 ± 3.1	0.9	61.8 ± 0.1	1.7	33.7 ± 3.5	2.0	28.6 ± 0.1	59.1 ± 3.1
**46**	1.5	41.8 ± 2.5	1.3	49.6 ± 7.0	2.2	29.5 ± 1.9	2.1	30.5 ± 2.3	64.9 ± 5.1
Doxorubicin	1.0	>100	1.0	>100	2.0	>100	1.0	>100	>100

## Data Availability

The data can be shared upon request.

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
