# Peer review of "Advancements in Synthetic Strategies and Biological Effects of Ciprofloxacin Derivatives: A Review"

_ijms, 2024, doi:10.3390/ijms25094919_

Round 1

Reviewer 1 Report

Comments and Suggestions for Authors

Minor revision is needed.

Reviewer 2 Report

Comments and Suggestions for Authors

This review examines recent advancements in the synthesis and pharmacological activities of ciprofloxacin derivatives. Ciprofloxacin, a widely used antibiotic belonging to the fluoroquinolone group, exhibits potent antibacterial effects by inhibiting bacterial topoisomerase enzymes. Researchers are exploring modifications to ciprofloxacin to enhance its therapeutic properties, including anticancer and antimicrobial activities.

Minor Comments:

1. It would be helpful to include a discussion on the challenges and limitations associated with the synthesis and clinical translation of ciprofloxacin derivatives.

2. Some of the references are out of date and the author may check and update them.

3. Some typos need to be corrected and the abbreviation should be cited in the first place.

Comments on the Quality of English Language

The quality of English language of this manuscript is clear and able to be understood.

Author Response

Response to Reviewer 2 Comments

  1. Summary

This review examines recent advancements in the synthesis and pharmacological activities of ciprofloxacin derivatives. Ciprofloxacin, a widely used antibiotic belonging to the fluoroquinolone group, exhibits potent antibacterial effects by inhibiting bacterial topoisomerase enzymes. Researchers are exploring modifications to ciprofloxacin to enhance its therapeutic properties, including anticancer and antimicrobial activities.

Minor Comments:

Comment 1: It would be helpful to include a discussion on the challenges and limitations associated with the synthesis and clinical translation of ciprofloxacin derivatives.

Response 1: The discussion on the challenges and limitations associated with the synthesis and clinical translation of ciprofloxacin derivatives has been added under the conclusion and future directions. Thank you for pointing this out.

Comment 2: Some of the references are out of date and the author may check and update them.

Response 2:  We reviewed and updated the manuscript's outdated replaceable references. Thank you for pointing this out. For example, We decided to retain some old references listed in Table 3 because there are few if any, recent studies on the anticancer activities of ciprofloxacin.

Comment 3: Some typos need to be corrected and the abbreviation should be cited in the first place.

Response 3: The typographical errors have been corrected and we provided the full term for every abbreviation when it first appears in the manuscript